# Sustained antidepressant effect of ketamine through NMDAR trapping in the LHb

Shuangshuang Ma[1,2,5], Min Chen[2,3,5], Yihao Jiang[2,3], Xinkuan Xiang[2], Shiqi Wang[2], Zuohang Wu[2], Shuo Li[2], Yihui Cui[2], Junying Wang[2], Yanqing Zhu[2], Yan Zhang[2], Huan Ma[2], Shumin Duan[2], Haohong Li[2], Yan Yang[2,3], Christopher J. Lingle[4] & Hailan Hu[1,2,3✉]

Ketamine, an *N*-methyl-ᴅ-aspartate receptor (NMDAR) antagonist[1], has revolutionized the treatment of depression because of its potent, rapid and sustained antidepressant effects[2–4]. Although the elimination half-life of ketamine is only 13 min in mice[5], its antidepressant activities can last for at least 24 h[6–9]. This large discrepancy poses an interesting basic biological question and has strong clinical implications. Here we demonstrate that after a single systemic injection, ketamine continues to suppress burst firing and block NMDARs in the lateral habenula (LHb) for up to 24 h. This long inhibition of NMDARs is not due to endocytosis but depends on the use-dependent trapping of ketamine in NMDARs. The rate of untrapping is regulated by neural activity. Harnessing the dynamic equilibrium of ketamine–NMDAR interactions by activating the LHb and opening local NMDARs at different plasma ketamine concentrations, we were able to either shorten or prolong the antidepressant effects of ketamine in vivo. These results provide new insights into the causal mechanisms of the sustained antidepressant effects of ketamine. The ability to modulate the duration of ketamine action based on the biophysical properties of ketamine–NMDAR interactions opens up new opportunities for the therapeutic use of ketamine.

Two characteristics of ketamine that make it attractive as a treatment for depression are its rapid onset and sustained activity[4]. A single intravenous infusion of a subanaesthetic dose of ketamine in patients with depression produces antidepressant and antisuicidal responses as rapidly as 1 h and the effects can last for days[2,3]. Regarding the rapid onset of action of ketamine, much progress has been made in understanding its mechanism[7–10]. One study showed that ketamine instantly blocks NMDAR-dependent bursting activity in the LHb[10], a signature neural activity in multiple animal models of depression[10–12]. As the anti-reward centre of the brain, the LHb inhibits the downstream aminergic reward centre[12–15] and is hyperactive in the depressive state[11,16–22]. Consequently, the rapid blockade of burst firing in the LHb by ketamine can potentially disinhibit the downstream dopaminergic and serotonergic neurons to quickly improve mood[10,23].

In comparison to the rapid effects of ketamine, the mechanisms that underlie its sustained antidepressant effects are less understood. The elimination half-life of ketamine is only approximately 3 h in humans[24] and 13 min in mice[5], yet its antidepressant activities can last for 3–14 days in humans[2,3,25] and for at least 24 h in mouse models of depression[6–9]. This result is in contrast to the anaesthetic effects of ketamine, which quickly wear off in a few hours[26], and to other classical antidepressants, which require daily intake to maintain an effective blood concentration[27]. As sustained efficacy will reduce repeated drug administration and unwanted side effects, how the antidepressant effects of ketamine can long outlast its plasma elimination is therefore not only an interesting basic biological question but also has important clinical implications.

Previously, the sustained effects of ketamine were attributed to long-term plastic mechanisms[6,8,28], especially new spine formation[7,29] or the ketamine metabolite (2*R*,6*R*)-hydroxynorketamine[9]. However, ketamine-induced spine growth in mice cannot be detected until 12 h after treatment[29], and the half-life of (2*R*,6*R*)-hydroxynorketamine is less than 30 min[9], not much longer than that of ketamine. Meanwhile, a much simpler mechanism that involves the direct blockade of NMDARs has not yet been explored. For this direct-blockade mechanism, the key question to answer is whether blockade of NMDARs continues to exist long after ketamine elimination in the brain. If yes, then the next question to address is how this prolonged blockade is achieved and whether it contributes to the sustained antidepressant effects of ketamine. In the current study, building on knowledge of the rapid effects of ketamine in the LHb, we use both in vitro and in vivo electrophysiological recordings to investigate whether persistent blockade of NMDARs and burst firing in the LHb may provide a foundation for the sustained antidepressant actions of this drug.

## Sustained behavioural effects of ketamine

We first mapped the antidepressant time course of a single systemic injection of ketamine in the chronic restraint stress (CRS) mouse model

[1]Department of Psychiatry and International Institutes of Medicine, The Fourth Affiliated Hospital, Zhejiang University School of Medicine, Yiwu, China. [2]Nanhu Brain–Computer Interface Institute, MOE Frontier Science Center for Brain Science and Brain–Machine Integration, State Key Laboratory of Brain–Machine Intelligence, New Cornerstone Science Laboratory, Zhejiang University, Hangzhou, China. [3]Department of Affiliated Mental Health Center and Hangzhou Seventh People's Hospital and School of Brain Science and Brain Medicine, Zhejiang University School of Medicine, Hangzhou, China. [4]Department of Anesthesiology, Washington University School of Medicine, St Louis, MO, USA. [5]These authors contributed equally: Shuangshuang Ma, Min Chen. ✉e-mail: huhailan@zju.edu.cn

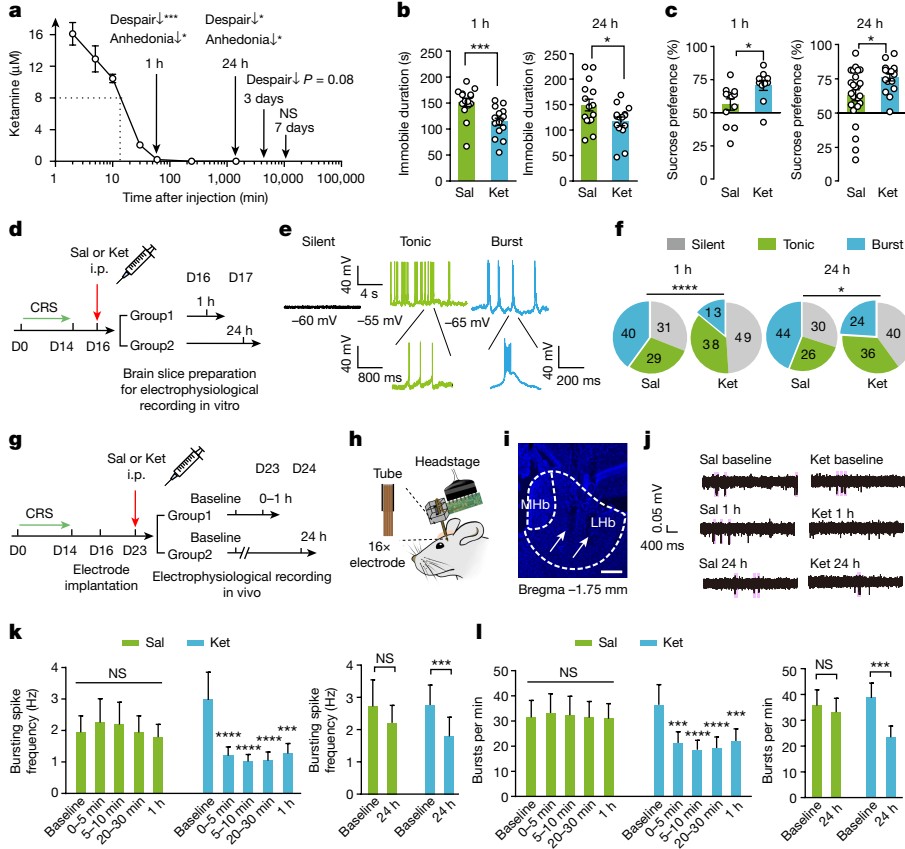

**Fig. 1 | A single injection of ketamine causes sustained antidepressant effects and prolonged suppression of LHb bursting activity. a**, Brain concentrations of ketamine after a single intraperitoneal (i.p.) injection (10 mg kg⁻¹) of ketamine in CRS mice, as measured by LC–MS/MS. Behavioural effects at different time points, as measured in **b** and **c** and Extended Data Fig. 1, are indicated. Dotted line indicates the half-life of ketamine, which is about 13 min. **b,c**, Behavioural effects at 1 h and 24 h after a single intraperitoneal injection of ketamine (Ket) or saline (Sal) in CRS mice in the FST (**b**) and the SPT (**c**). **d**, Experimental paradigm for LHb slice recording after intraperitoneal injection of ketamine in CRS mice. D, day. **e**, Representative traces showing spontaneous activity of three LHb neuron types: burst firing, tonic firing and silent. **f**, Pie charts illustrating the per cent abundance of the three types of LHb neurons in CRS mice at different time points after saline or ketamine injection.

**g**, Experimental paradigm for in vivo recording after a single injection of ketamine in CRS mice. **h**, Illustration of in vivo single-unit recording in the LHb. **i**, An example recording site stained with DAPI. White dotted lines demarcate the medial habenula (MHb) and the LHb. White arrows indicate the electrode tracks. Scale bar, 100 μm. **j**, Example traces showing in vivo neuronal activity recorded in CRS mice before, 1 h and 24 h after saline or ketamine injection. Bursts (pink shades) are identified using the inter-spike interval method (Methods). **k,l**, Bar graphs illustrating bursting spike frequency (**k**) and bursts per minute (**l**) in CRS mice at different time points after saline or ketamine injection. Note that the data are not normally distributed so nonparametric tests were used for statistics. *$P < 0.05$, **$P < 0.01$, ***$P < 0.001$, ****$P < 0.0001$. NS, not significant. Error bars indicate the s.e.m. (see Supplementary Table 1 for statistical analyses and $n$ numbers).

of depression. After exposure to 14 days of restraint stress, mice were injected with ketamine (intraperitoneal, 10 mg kg⁻¹). At different time points (1 h, 24 h, 3 days and 7 days) after ketamine injection, we measured both plasma and brain concentrations of ketamine using liquid chromatography–tandem mass spectrometry (LC–MS/MS) and tested mice for depressive-like behaviours (Fig. 1a and Extended Data Fig. 1a,b). Depressive-like behaviours were measured using the forced swim test (FST), which models behavioural despair, and the sucrose preference test (SPT), which models anhedonia or the inability to feel pleasure. Compared with the saline-treated group, after ketamine injection, the immobility time in the FST was significantly decreased and the preference for sucrose water in the SPT was significantly increased at 1 h ($P = 0.0005$ for FST, and $P = 0.011$ for SPT, Mann–Whitney test), and at 24 h ($P = 0.033$ for FST, and $P = 0.02$ for SPT, unpaired $t$-test) (Fig. 1b,c). At 3 days after ketamine injection, there was an antidepressant trend in the FST ($P = 0.081$, unpaired $t$-test) but not the SPT (Extended Data Fig. 1c,d). At 7 days after injection, the antidepressant effects became nonsignificant (Extended Data Fig. 1c,d). Notably, LC–MS/MS showed that both the plasma and brain concentrations of ketamine rapidly dropped after injection (Fig. 1a and Extended Data Fig. 1b).

By 1 h, it had reduced to a level (0.23 μM in the brain) too low to inhibit NMDARs (half-maximum inhibitory concentration (IC₅₀) of about 5 μM in the presence of magnesium[30]); however, the antidepressant effects remained significant until 24 h (Fig. 1b,c). These paradoxical results confirmed previous reports on the short half-life and sustained behavioural effects of ketamine[6–9], which calls for a mechanistic explanation.

## Sustained LHb suppression by ketamine

As the blockade of LHb burst firing mediates the rapid antidepressant effects of ketamine[10], we wished to examine whether it may also underlie the sustained effects of ketamine. We first examined how long a single intraperitoneal injection of ketamine can suppress the bursting activity of LHb neurons (Fig. 1d). Spontaneous neuronal activity at resting conditions was recorded from coronal LHb brain slices taken from CRS mice at 1 h, 24 h or 3 days after intraperitoneal injection of ketamine (Fig. 1d). As previously shown[10,12,23,31], LHb neurons were intrinsically active and could be categorized into silent, tonic firing and burst firing types (Fig. 1e). The proportion of burst firing neurons was significantly higher in CRS mice compared with naive mice[10] (Fig. 1f and Extended

Data Fig. 2a). In slices prepared 1 h after ketamine injection, the percentage of burst firing neurons reduced from 40% in the saline-treated group to 13% in the ketamine-treated group ($P < 0.0001$, Chi-square test; Fig. 1f). Notably, in slices prepared 24 h after ketamine injection, the percentage of burst firing neurons was still significantly reduced (44% in the saline-treated group compared with 24% in the ketamine-treated group, $P = 0.012$, Chi-square test; Fig. 1f). By 3 days after ketamine injection, the difference was no longer significant (40% in the saline-treated group compared with 33% in the ketamine-treated group, $P = 0.48$; Extended Data Fig. 2b). The bursting spike frequency (defined as the number of bursting spikes per second) and bursts per minute (defined as the number of bursts per minute) of recorded LHb neurons showed a similar trend in change. That is, significantly suppressed at 1 h and 24 h but not at 3 days after ketamine injection (Extended Data Fig. 2c–e).

We next tested whether the sustained suppression of LHb bursting induced by a single ketamine injection also occurs in vivo. To that end, we performed single-unit recording in the LHb of freely moving CRS mice before and at different time points after intraperitoneal injection of ketamine (Fig. 1g–l and Extended Data Fig. 2f–k). Suppression of LHb bursting activity, assessed by measuring the bursting spike frequency and bursts per minute, was significant as early as the first 5 min, maximized at 5–10 min and persisted up to 24 h after ketamine injection (Fig. 1k,l). By 3 days after ketamine injection, the suppression of LHb bursting activity was not significant (Extended Data Fig. 2h,i). By comparison, saline treatment produced no alteration of LHb bursting activity during long-term in vivo recording in CRS mice (Fig. 1k,l and Extended Data Fig. 2h–k).

Further analysis revealed that the bursting spike frequency of LHb neurons had a bimodal distribution, and ketamine preferentially affected the population with a higher basal bursting spike frequency (Extended Data Fig. 3). In particular, even though only 27% (26 out of 97) LHb neurons had a bursting spike frequency larger than 2 Hz (>2 Hz group) (Extended Data Fig. 3a), the majority (for example, 76.9% at 1 h) of ketamine-inhibited neurons was in this >2 Hz group (Extended Data Fig. 3b–d). Such activity-dependent inhibition is consistent with the use-dependent nature of blockade of ketamine. That is, blocking the NMDAR channel only when it is in the open state[32]. Collectively, both in vitro and in vivo recording data demonstrated that a single systemic injection of ketamine in depressive-like mice elicits sustained inhibition of LHb bursting activity in a time course that parallels its behavioural effects.

## Prolonged LHb NMDAR blockade by ketamine

To understand why ketamine can cause sustained suppression of LHb bursting, and because LHb bursting activity depends on NMDARs[10], we next measured in LHb brain slices how long ketamine can continue to block NMDAR currents after a single intraperitoneal injection (Fig. 2 and Extended Data Fig. 4). Sagittal LHb brain slices of CRS mice were prepared at 1 h, 24 h or 3 days after mice were injected with ketamine (intraperitoneal, 10 mg kg⁻¹). A stimulating electrode close to the input stria medullaris fibre was placed on the slices, and whole-cell patch-clamp was performed to record evoked excitatory postsynaptic currents (eEPSCs) (Fig. 2a,b). In LHb slices from both saline-treated and ketamine-treated mice, we isolated AMPAR-mediated and NMDAR-mediated excitatory postsynaptic currents (AMPAR-eEPSCs and NMDAR-eEPSCs, respectively) based on their temporal characteristics (Methods, Fig. 2c,i and Extended Data Fig. 4a). In LHb slices prepared 1 h after ketamine injection, there was a significant decrease in the ratio of NMDAR-eEPSCs and AMPAR-eEPSCs ($0.30 \pm 0.06$ for the saline-treated group, and $0.03 \pm 0.01$ for the ketamine-treated group, $P < 0.0001$, Mann–Whitney test; Fig. 2d). To further distinguish whether this reduction was due to a change in NMDAR-eEPSCs or AMPAR-eEPSCs, we analysed the input–output curves of NMDAR-eEPSCs and AMPAR-eEPSCs,

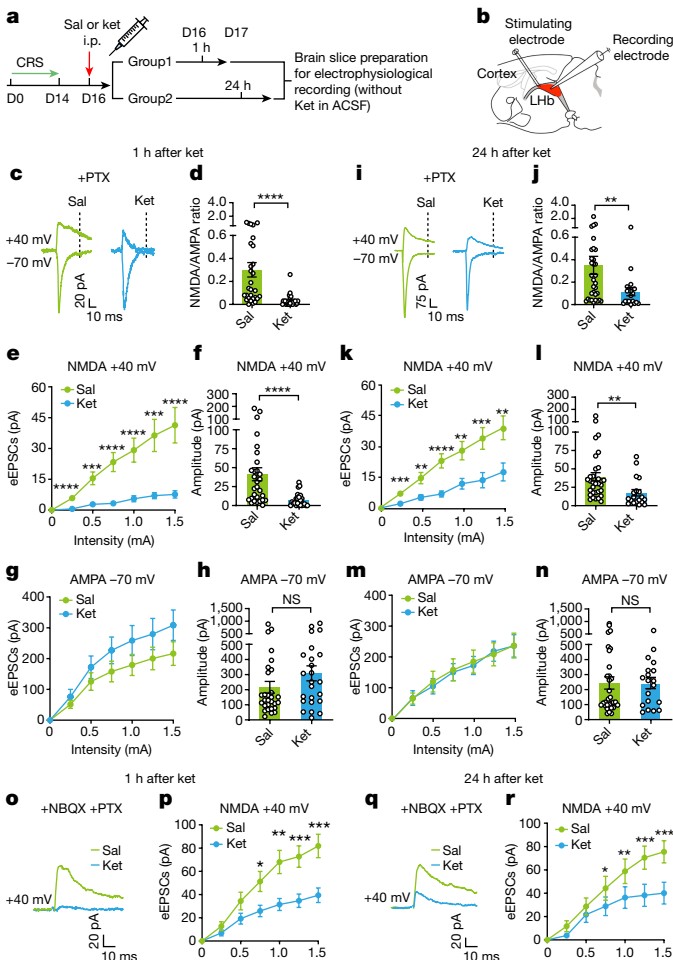

**Fig. 2 | A single injection of ketamine in CRS mice causes prolonged inhibition of NMDAR currents in the LHb. a**, Experimental paradigm for slice recording after intraperitoneal injection of ketamine (10 mg kg⁻¹) in CRS mice. **b**, Schematic of whole-cell recording of evoked synaptic responses in sagittal LHb slices. **c–n**, Data from temporally isolated AMPAR-eEPSCs and NMDAR-eEPSCs. **c,i**, Example traces of evoked AMPAR-eEPSCs (–70 mV, measured at the peak) and NMDAR-eEPSCs (+40 mV, measured at 35 ms after stimulation, dotted line) in LHb neurons in the presence of picrotoxin (PTX) at 1 h (**c**) and 24 h (**i**) after intraperitoneal injection of saline or ketamine in CRS mice. **d,j**, Ratios of NMDAR-eEPSCs and AMPAR-eEPSCs (recorded at 1.5 mA stimulation intensity) at 1 h (**d**) and 24 h (**j**) after intraperitoneal injection of saline or ketamine in CRS mice. **e,g,k,m**, Stimulus–response (input–output) curves of NMDAR-eEPSCs (**e,k**) and AMPAR-eEPSCs (**g,m**) of LHb neurons at 1 h (**e,g**) and 24 h (**k,m**) after intraperitoneal injection of saline or ketamine in CRS mice. **f,h,l,n**, Bar graphs of NMDAR-eEPSCs (**f,l**) and AMPAR-eEPSCs (**h,n**) recorded at 1.5 mA stimulation intensity at 1 h (**f,h**) and 24 h (**l,n**) after injection of saline or ketamine in CRS mice. **o–r**, Data from pharmacologically isolated pure NMDAR-eEPSCs. **o,q**, Example traces of evoked NMDAR-eEPSCs (+40 mV, measured at the peak) of LHb neurons in the presence of picrotoxin and NBQX at 1 h (**o**) and 24 h (**q**) after injection of saline or ketamine. **p,r**, Stimulus–response (input–output) curves of NMDAR-eEPSCs (isolated by application of picrotoxin and NBQX under voltage clamp at +40 mV) of LHb neurons at 1 h (**p**) and 24 h (**r**) after injection of saline or ketamine. Error bars indicate the s.e.m. (see Supplementary Table 1 for statistical analyses and *n* numbers).

respectively. NMDAR-eEPSCs of the ketamine-treated group showed strongly reduced amplitudes across a range of stimulation intensities compared with the saline-treated group (Fig. 2e). For example, the amplitudes of NMDAR-eEPSCs recorded at 1.5 mA stimulation intensity were $41.3 \pm 8.6$ pA from the saline-treated group (*n* = 29 in 3 mice) and $7.6 \pm 1.8$ pA from the ketamine-treated group (*n* = 24 in 3 mice,

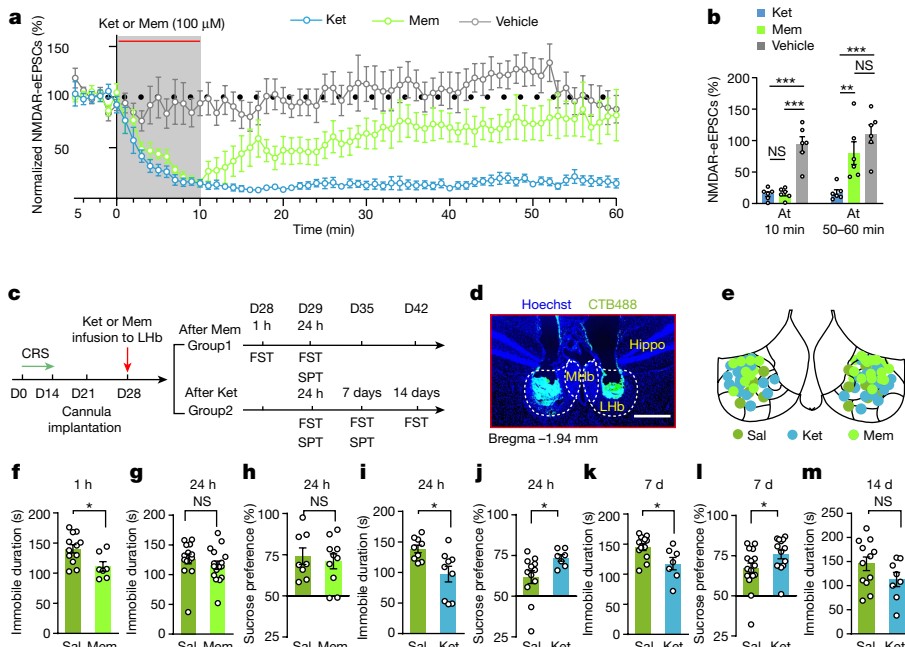

**Fig. 3 | Prolonged blockade of LHb NMDARs after ketamine wash off.**
**a**, NMDAR-eEPSCs (normalized by baseline) during incubation and washout of vehicle, memantine (Mem; 100 μM) or ketamine (100 μM). NMDAR-eEPSCs were isolated through the application of picrotoxin and NBQX in $Mg^{2+}$-free ACSF under voltage clamp at −70 mV. **b**, Bar graphs showing NMDAR-eEPSCs at the end of the 10-min perfusion period (left) and at 50–60 min (right). **c**, Experimental paradigm for behavioural testing after local bilateral infusion of ketamine (100 mM, 0.1 μl each side) or memantine (100 mM, 0.1 μl each side) into the LHb of CRS mice. **d**, Illustration of bilateral implantation of cannula in

the LHb of CRS mice. CTB, cholera toxin subunit B; Hippo, hippocampus. White dashed lines demarcate the MHb and the LHb. Scale bar, 500 μm. **e**, Infusion sites of drugs verified by CTB. **f**–**h**, Behavioural effects at 1 h and 24 h after local bilateral infusion of memantine into the LHb of CRS mice in the FST (**f**,**g**) and SPT (**h**). **i**–**l**, Behavioural effects at 24 h (**i**,**j**) and 7 days (**k**,**l**) after local bilateral infusion of ketamine into the LHb of CRS mice in the FST (**i**,**k**) and SPT (**j**,**l**). **m**, Behavioural effects at 14 days after local bilateral infusion of ketamine into the LHb of CRS mice in the FST. Error bars indicate the s.e.m. (see Supplementary Table 1 for statistical analyses and *n* numbers).

*P* < 0.0001, Mann–Whitney test; Fig. 2f). The prevalence of neurons with large-amplitude NMDAR-eEPSCs (>10 pA) was notably lower in the ketamine-treated group (69% for the saline-treated group, and 29% for the ketamine-treated group, *P* < 0.0001, Chi-square test; Extended Data Fig. 4b). By contrast, the input–output curves and the amplitudes of AMPAR-eEPSCs were not significantly different between the treatment groups (*P* = 0.11, Mann–Whitney test; Fig. 2g,h).

In LHb slices prepared 24 h after ketamine injection, strong inhibition of NMDARs (under 1.5 mA stimulation, 39.09 ± 5.77 pA for the saline-treated group compared with 17.58 ± 4.41 pA for the ketamine-treated group, *P* = 0.001, Mann–Whitney test; Fig. 2k–n) and decreased NMDA/AMPA ratios (0.35 ± 0.08 for the saline-treated group, and 0.12 ± 0.04 for the ketamine-treated group, *P* = 0.004, Mann–Whitney test; Fig. 2j) persisted. By 3 days after ketamine injection, the blockade was no longer significant (Extended Data Fig. 4d–f). The levels of NMDAR-eEPSC inhibition (see Methods for calculation) in the LHb at 1 h, 24 h and 3 days after ketamine injection were 81.5 ± 4.4%, 55.0 ± 11.3% and −1.2 ± 25.8%, respectively (Extended Data Fig. 4g). Again, these results paralleled the time course of the behavioural effects of ketamine. To further confirm the persistent blockade of NMDARs by ketamine, we pharmacologically isolated pure NMDAR-eEPSCs in the LHb of CRS mice in the presence of both the GABA_AR blocker picrotoxin and the AMPAR blocker NBQX. The pharmacologically isolated NMDAR-eEPSCs also showed significant blockade at 24 h after ketamine intraperitoneal injection (Fig. 2o–r).

## NMDAR blockade persists after ketamine washout

We next investigated the potential mechanism by which ketamine may continue to block NMDARs long after its elimination in the brain. It is of interest to note that in the slice recording paradigm in Fig. 2, even

for the 1 h time point, LHb slices had been incubated and perfused in ketamine-free artificial cerebrospinal fluid (ACSF) solution for hours, and the residual ketamine in the tissue from intraperitoneal injection should have been washed off by the time of recording. The fact that NMDARs still showed strong inhibition suggests that ketamine may be trapped in the NMDAR channel, which prevents it from being cleared. Indeed, as an open channel blocker, after binding to the channel, ketamine can be trapped in the channel pore, getting released only when the channel is open again[33,34]. To test the trapping hypothesis more directly, we continuously monitored NMDAR-eEPSC responses in LHb brain slices while washing in and out ketamine or memantine, another pore-blocking-type NMDAR inhibitor with a similar affinity but faster off-rate compared with ketamine[35] (Fig. 3a). NMDAR-eEPSCs were pharmacologically isolated at −70 mV (in the absence of magnesium to remove the magnesium-mediated blockade of NMDARs; Methods and Fig. 3a). After a 5-min recording of a stable baseline, ketamine (100 μM) or memantine (100 μM) was perfused into the recording ACSF and then washed out 10 min later, and the recovery of NMDAR-eEPSCs in the next 50 min was observed (Fig. 3a). As control, NMDAR-eEPSCs did not change in the vehicle-treated group during this entire length of recording (Fig. 3a), and the input resistance of recorded neurons remained constant throughout the recording period (Extended Data Fig. 5), which indicated stable recording. At the end of the 10-min drug wash-in period, NMDAR-eEPSCs showed an equivalent level of strong reduction by the two drugs (85.1 ± 3.7% reduction for ketamine, *n* = 6 cells; 84.3 ± 3.5% reduction for memantine, *n* = 6 cells; Fig. 3b). After drug washout, the memantine-blocked NMDAR-eEPSCs quickly recovered, and by 50 min, only 19.9 ± 18.2% reduction remained (Fig. 3b). By contrast, the ketamine-blocked NMDAR-eEPSCs continued to be blocked after ketamine washout, and by 50 min, the reduction in NMDAR-eEPSCs currents was still as large as 82.7 ± 4.7% (Fig. 3b).

As the treatment-relevant peak concentration of ketamine in the brain is around 16 μM (Fig. 1a), we repeated the above washout experiment using a treatment-relevant concentration of 10 μM ketamine (Fig. 4). The NMDAR-eEPSC responses decayed slower during the 10-min perfusion period for the 10 μM ketamine condition than for the 100 μM ketamine condition. However, significant blockade of NMDAR responses (73.8 ± 7.1% reduction, $n$ = 6 cells) still lasted at least for 50 min (Fig. 4a, black). To eliminate the possibility of NMDAR endocytosis, we introduced Dyngo-4a, an NMDAR endocytosis inhibitor, which can effectively suppress low-frequency stimulation (LFS)-induced endocytosis of hippocampal NMDARs[36] (Extended Data Fig. 6a). In the presence of Dyngo-4a, LHb NMDAR-eEPSCs still did not recover after either 100 μM or 10 μM ketamine washout (Extended Data Fig. 6b–e), which suggested that the persistent suppression of NMDAR-eEPSCs in the LHb after ketamine washout is not due to endocytosis. Together, these results suggest that LHb NMDARs can be continuously blocked long after the clearance of ketamine in plasma.

Notably, when we analysed individual neurons recorded (Figs. 3a and 4a), a reverse correlation was observed between the level of initial NMDAR blockade by ketamine and the level of recovery. That is, neurons with a higher percentage of blockade after ketamine wash-in tended to have a lower level of recovery at the end of the 50-min washout period ($R^2$ = 0.50, $P$ = 0.01, linear regression; Extended Data Fig. 7). This result suggests that the more complete the blockade, the slower recovery of the NMDAR currents. This reverse correlation has also been reported in hippocampal neurons, where it reflects the contribution of a reservoir pool of extrasynaptic NMDARs in recovery[37].

## Local ketamine produces extended antidepressant effects

If prolonged blockade of NMDARs in the LHb contributes causally to the sustained antidepressant effects of ketamine, we proposed that local infusion of ketamine into the LHb should mimic the sustained effects of systemic ketamine injection. To test this hypothesis, we performed bilateral infusion of either ketamine (100 mM, 0.1 μl each side) or memantine (100 mM, 0.1 μl each side) into the LHb of CRS mice through a dual-guide cannula and measured depressive-like behaviours days later (Fig. 3c–m). Consistent with its rapid washout in LHb brain slice recordings (Fig. 3a,b), infusion of memantine did not produce antidepressant effects in CRS mice at 24 h (Fig. 3f–h). By contrast, lasting antidepressant effects were present in both the FST and SPT at 24 h, and even 7 days, after ketamine infusion into the LHb (Fig. 3i–m). We previously found that local infusion of ketamine into the LHb causes antidepressant effects as early as 1 h[10]. Therefore, local infusion of ketamine in the LHb is sufficient to recapitulate both the rapid (1 h) and sustained (24 h) antidepressant effects of systematic ketamine injection.

## Neural activity untraps ketamine

We next reasoned that if trapping of ketamine in the channel pore accounts for the prolonged blockade of NMDARs, untrapping it should quickly bring back NMDAR responses in the brain slice washout experiments (Fig. 4a). One way to rapidly release a voltage-dependent trapping blocker is to open the NMDAR channel again through agonist binding and accelerate its release through voltage depolarization[38,39]. Therefore we designed a 'kick-off' protocol, whereby presynaptic 1 Hz electrical stimulation (to release glutamate) was paired with postsynaptic current injection (to depolarize neurons, 3 s at +10 mV per min)[39]. At 10 min after ketamine washout, we introduced two such kick-off sessions (5 min per session, 30 spikes total) of neural activity, and, as predicted, NMDAR-eEPSCs recovered to 80.0 ± 11.3% of baseline level by 50 min (Fig. 4). By contrast, a protocol with 1 Hz presynaptic stimulation alone did not result in such recovery (Fig. 4). The kick-off protocol itself did not cause any sustained change in the NMDAR-eEPSCs of LHb

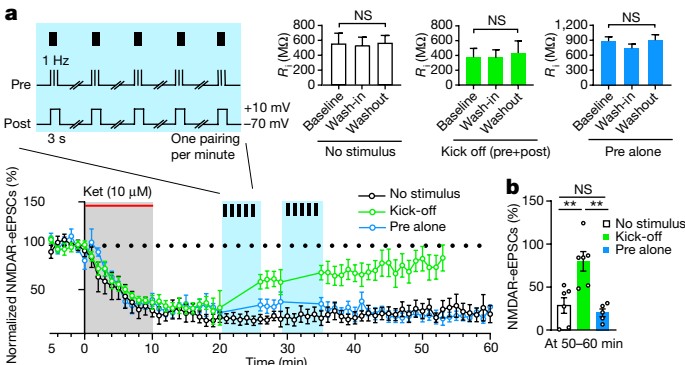

**Fig. 4 | Neural activity untraps ketamine from NMDARs. a**, Top left, schematic of a kick-off protocol used to untrap ketamine, including five 3-s pairings of presynaptic stimulation (1 Hz) with postsynaptic depolarization (to +10 mV) delivered over a 5-min period. Bottom left, NMDAR-eEPSCs (normalized by baseline) during incubation and washout of ketamine (10 μM) under three conditions: no stimulus, no extra stimulation; kick-off, two blocks of kick-off activity given after ketamine washout; pre alone, two blocks of presynaptic stimulation alone without postsynaptic depolarization given after ketamine washout. Top right, input resistance of recorded neurons during baseline, wash-in (at 10 min) and washout (at 50 min) periods for three different conditions. **b**, Bar graphs showing NMDAR-eEPSCs at 50–60 min. Error bars indicate the s.e.m. (see Supplementary Table 1 for statistical analyses and $n$ numbers).

neurons without ketamine treatment (Extended Data Fig. 8), which suggested that the recovery of NMDAR-eEPSCs was not due to long-term potentiation (LTP).

## LHb activation at low [Ket] shortens antidepression

The above-described in vitro experiments indicated the potential of kicking-off ketamine from NMDARs in LHb neurons through neural activity. Subsequently, we explored whether this understanding combined with the pharmacokinetics of ketamine could be harnessed to modulate the duration of its behavioural effects in vivo (Fig. 5). In contrast to the in vitro recording condition, magnesium is present in vivo; and it is not possible to voltage-clamp neurons in the in vivo environment. Instead, to induce LHb neural activity and to open NMDARs in vivo, we stimulated a major LHb input pathway, the lateral hypothalamus (LH)[40–43], to induce LHb burst firing. Using this strategy, short-interval spikes can produce concomitant glutamate release and the postsynaptic depolarization required for NMDAR opening[42] (Extended Data Fig. 9a–c). To stimulate the LH input to the LHb, we expressed AAV-ChrimsonR-tdTomato in the LH of CRS mice, and through bilateral optic fibres, stimulated the LH terminals in the LHb (635 nm, 40 Hz, 2-ms pulse, 250 μW; Methods and Fig. 5a,b). As LH–LHb stimulation causes aversive effects[40–42], we tested the effectiveness of the 40-Hz protocol to induce real-time place aversion (RTPA; Fig. 5c) in every animal that underwent the behavioural protocol (Fig. 5f,j).

As a use-dependent open channel blocker, the interaction of ketamine with NMDARs depends on the open state of the channel, which we could manipulate using the above-mentioned optogenetic stimulation protocol. When NMDARs are open and accessible to ketamine entry, according to the dynamic equilibrium, the binding and unbinding of ketamine will be strongly regulated by its ambient concentration. That is, when the ambient ketamine concentration is lower than the dissociation constant ($K_d$), the interaction leads to more unbinding (Fig. 5d,e). Conversely, when the ambient ketamine concentration is higher than the $K_d$, it leads to more binding (Fig. 5d,i). Therefore, for the in vivo kick-off or unbinding experiment, we tried to induce NMDAR opening at a low ambient ketamine concentration by stimulating the

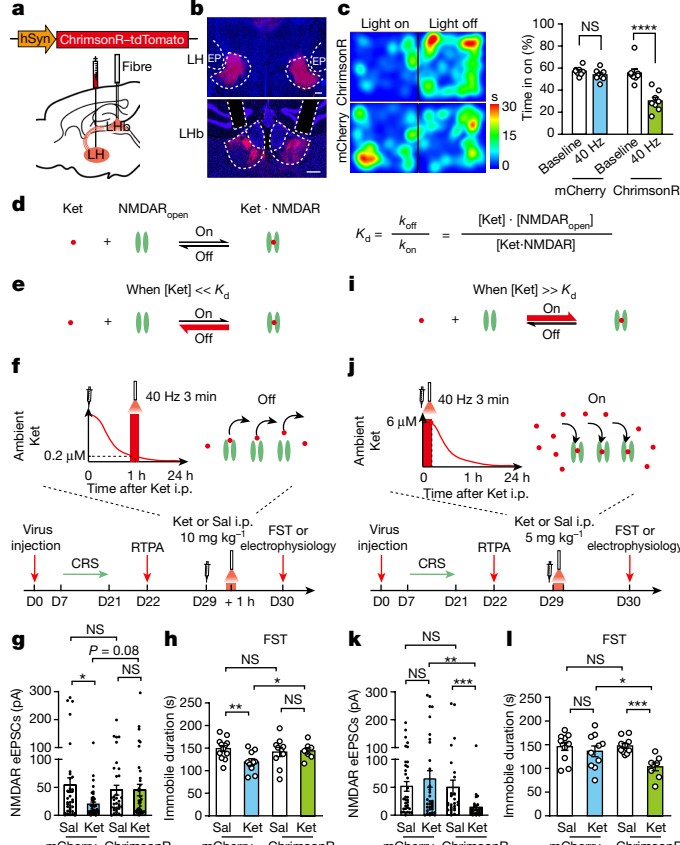

**Fig. 5 | Activation of the LHb input pathway at different ambient ketamine levels bidirectionally regulates the sustained antidepressant effects of ketamine. a**, Schematic of viral construct, viral injection in the LH and optic fibre implantation in the LHb of CRS mice. **b**, An example image showing bilateral viral injection sites in the LH (top), viral expression in axon terminals in the LHb and canular sites for optic fibre implantation (bottom). EP, entopeduncular nucleus. Scale bar, 200 μm. **c**, RTPA induced by constant 40 Hz light stimulation of the LH–LHb axon terminals. Left, representative heatmaps. Right, percentage of time spent in the light-stimulated chamber. **d**, Equation of the dynamic equilibrium for the ketamine–NMDAR interaction. **e**, When the ambient ketamine concentration ([Ket]) is lower than the $K_d$, the interaction between ketamine and NMDAR leads to more unbinding. **f**, Experimental paradigm of stimulating the LH–LHb at low ketamine concentrations to untrap ketamine. After CRS, RTPA was conducted to confirm the effectiveness of LH–LHb stimulation. One hour after a ketamine injection (intraperitoneal, 10 mg kg⁻¹), stimulation was delivered to LH–LHb terminals when the brain concentration of ketamine has dropped to 0.23 μM. Electrophysiological recording or behavioural testing was conducted 23 h later. **g**,**h**, NMDAR-eEPSCs of LHb neurons (**g**) and behavioural effects (**h**) at 24 h after ketamine or saline injection from the experiment in **f**. **i**, When the ambient ketamine concentration is higher than the $K_d$, the interaction between ketamine and NMDAR leads to more binding. **j**, Experimental paradigm of stimulating the LH–LHb at high ketamine concentrations to block more NMDARs. Immediately after a ketamine injection (intraperitoneal, 5 mg kg⁻¹), stimulation was delivered to LH–LHb terminals, when the brain concentration of ketamine is above 6 μM. Electrophysiological recording or behavioural testing was conducted 24 h later. **k**,**l**, NMDAR-eEPSCs of LHb neurons (**k**) and behavioural effects (**l**) at 24 h after ketamine or saline injection from the experiment in **j**. Error bars indicate the s.e.m. (see Supplementary Table 1 for statistical analyses and *n* numbers).

LH–LHb terminals (40 Hz, 3 min) at 1 h after ketamine injection, when its ambient concentration had dropped to 0.23 μM (Fig. 1a), several fold below the $K_d$ (about 0.8 μM)[35] (Fig. 5e,f and Extended Data Fig. 9d). We first examined the kick-off effect by recording NMDAR-eEPSCs of LHb brain slices prepared immediately after optical stimulation (Extended Data Fig. 9d–g). Compared with the non-stimulated group, there was a

significant increase in NMDAR-eEPSCs, which suggested that ketamine had been released from the NMDARs through the in vivo LH–LHb optical stimulation method (Extended Data Fig. 9d–g). To test the effects of such a kick-off protocol on sustaining antidepressant effects, we measured both the LHb–NMDAR responses and behavioural effects of LH stimulation at 24 h after ketamine administration. In control mCherry-expressing mice, which do not have light-induced activity in the LHb, ketamine still suppressed NMDAR-eEPSCs and reduced immobility in the FST 24 h after intraperitoneal injection (Fig. 5g,h). In ChrimsonR-expressing mice, which have light-induced activity in the LHb, ketamine no longer had such effects at the 24 h time point (Fig. 5g,h). These results suggest that in vivo induced activity in the LHb at a time point with low ambient ketamine concentrations eliminates the sustained effects of ketamine on both LHb NMDARs and antidepressant behaviours.

## LHb activation at high [Ket] extends antidepression

For clinical relevance, it would be desirable to achieve the opposite effects to those described above. That is, to extend the duration of the antidepressant effects of ketamine or to achieve the same duration with a lower dosage. When the dose of ketamine intraperitoneal injection was reduced from 10 to 5 mg kg⁻¹, the antidepressant effects were still observed at 1 h[28] but no longer at 24 h[44] (Fig. 5l). One explanation for this shortened effect is that at this reduced dose, more of the blocked LHb NMDARs may have recovered by 24 h (Fig. 5k). Given the reverse correlation between the level of initial ketamine blockade and the level of recovery (Extended Data Fig. 7), we reasoned that if we could block more LHb NMDARs after ketamine treatment, we may slow down its recovery and extend the therapeutic effects of ketamine at 5 mg kg⁻¹. To increase ketamine blockade, we chose to stimulate the LH–LHb pathway to open more NMDARs at a time when the ambient ketamine level is high (Fig. 5i,j). We injected CRS mice with ketamine at 5 mg kg⁻¹ and then stimulated the LH–LHb pathway (635 nm, 3 min, 40 Hz) immediately after ketamine injection, when the ambient ketamine level was greater than 6 μM (Extended Data Fig. 10), several fold above the $K_d$ (Fig. 5j). Compared with the mCherry-expressing group, the ChrimsonR-expressing group was anticipated to show more ketamine NMDAR binding after light stimulation and a slower recovery of NMDAR responses (Fig. 5j). At 24 h after injection with 5 mg kg⁻¹ of ketamine, mCherry-expressing mice did not show any differences compared with saline-treated mice. By contrast, in the ChrimsonR-expressing mice, ketamine-treated mice showed more suppressed NMDAR responses and reduced immobility in the FST (Fig. 5k,l). There were also significant differences between the ketamine-treated ChrimsonR-expressing group and the mCherry-expressing group (Fig. 5k, l). These results suggest that in vivo induced activity in the LH–LHb pathway at a time point of high ambient ketamine level extended its effects on both inhibiting LHb NMDARs and reducing depression-like behaviours.

## Discussion

In the current work, we examined whether the sustained antidepressant efficacy of ketamine is a direct consequence of its specific pharmacodynamics. We demonstrated that even though ketamine concentrations decay rapidly in the plasma and the brain, and fall below the IC₅₀ within 30 min, ketamine continues to block NMDARs and suppress burst firing in the LHb for up to 24 h in mice (Figs. 1 and 2). The sustained inhibition of NMDARs by ketamine is not due to endocytosis and can be displaced by neural activity (Figs. 3 and 4). By activating the LH–LHb circuit at different time points after ketamine treatment, we were able to either shorten or extend the antidepressant effects of ketamine (Fig. 5). These findings support a hypothesis whereby long-term blockade of NMDARs within the LHb region mediates the sustained antidepressant effects of this drug, at least within 24 h. Beyond the 24 h time point, it is possible

that neural plasticity and other secondary mechanisms may account for the sustained effects at an even longer time scale[6–9,28,29]. Nevertheless, our study illustrates a case whereby a biophysical channel block mechanism extended to long time periods through in vivo physiological parameters may explain an important therapeutic function.

At the molecular level, it was notable that a single dose of ketamine blocked NMDAR channels for as long as 24 h. This duration is orders-of-magnitude longer than its reported dissociation time (5–13 s)[34], which was measured in vitro in steady-state NMDARs in dissociated cells in the constant presence of saturating amount of agonists (10 μM glycine, 10 μM NMDA)[45]. In the physiological synaptic environment, however, following a single stimulation, postsynaptic NMDARs are transiently exposed to glutamate for only 1–2 ms[45], and the duration of channel opening is less than 10 ms[46]. In addition, without stimulation, the intrinsic open probability of NMDARs in synapses is negligible ($P_o = 0.04$)[47]. Therefore, it is plausible that under low-level neural activity in vivo, the trapped ketamine cannot be efficiently released from NMDARs. Another factor that could contribute to the extended action time of ketamine in vivo is delayed diffusion[48]. Unlike transmitters that have transporter-based clearance mechanisms, ketamine could rebind to unbound NMDARs multiple times as it is released and diffuses out of synaptic clefts or extracellular spaces where astrocytic endfeet tightly wrap around neurons[49]. Given the different results obtained in the washout experiments between memantine and ketamine (Fig. 3), such lateral diffusion seems unlikely to be the major explanation for the sustained effects of ketamine, but may nevertheless contribute to its extended action time. As a consequence of the above factors, the apparent $k_{off}$ of ketamine in vivo can be much longer than previously measured in-solution $k_{off}$ values. Therefore once bound, a significant amount of ketamine is trapped, isolating it from metabolic degradation and sustaining its antidepressant effects.

Currently, there is an intense debate about whether the antidepressant effects of ketamine are mediated by NMDARs at all[9,25,50]. One major argument against an NMDAR-based mechanism is that unlike ketamine, many other NMDAR inhibitors failed to show comparable clinical efficacy[25]. However, this notion was directly challenged by a recent phase 3 clinical trial demonstrating that in conjunction with a compound to slow down its metabolism, AXS-05, another pore-trapping-type NMDAR inhibitor, effectively improved the condition of patients with depression[51]. The success of AXS-05 indicated that the root cause of failure for other NMDAR antagonists is not the target but rather the properties of the compounds themselves. Different NMDAR inhibitors have distinct inhibition mechanisms (for example, competitive binding or allosteric binding compared to pore trapping)[52,53], pharmacokinetics and pharmacodynamics. As illustrated in this study, even an NMDAR inhibitor that shares a similar trapping mechanism and a similar binding affinity as ketamine but with a faster $k_{off}$—memantine[35]—can have less optimal and less lasting antidepressant effects[54,55] (Fig. 3). These results demonstrate that the distinct pharmaceutical features of ketamine are crucial for its antidepressant effects, and that optimization of these properties is a promising new direction for developing new antidepressant treatments.

Our discoveries have several direct clinical implications. First, local application of ketamine in the LHb produced even more long-lasting antidepressant effects than systemic application, which were 7 days in mice (Fig. 3k,l) and 14 days in rats (data not shown). This effect may be explained by the fact that by locally targeting the LHb region, we were able to administrate ketamine at a much higher concentration (for example in the millimolar range), which ensured that a larger fraction of NMDARs were blocked and, therefore, a longer duration of NMDAR inhibition (Extended Data Fig. 7). This high local concentration cannot be achieved through systemic administration because of the well-known side effects of ketamine on peripheral organs[56]. Clinically, local administration at the LHb could be achieved possibly through liposome delivery combined with focused ultrasound[57] or before a deep-brain stimulation

protocol in the region[58]. Second, our data suggested that the effective duration of ketamine treatment depends on the regulation and interference by local neural activity. That is, activating the LHb at low ambient ketamine level shortens its antidepressant effects, whereas activating the LHb at high ambient ketamine level extends these effects (Fig. 5). As aversive stimuli can effectively activate the LHb under physiological conditions[12], these data suggest two appealing, clinically testable, strategies to extend the efficacy of ketamine. One is to apply ketamine during a depressive episode or to activate the LHb during ketamine administration by moderate aversive stimuli such as noise[59] or other acute stressor[12] to open more LHb NMDARs for more ketamine trapping. The other strategy is to minimize negative emotional episodes after ketamine treatment to slow down ketamine untrapping.

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

## Methods

### Animals

Male adult (8–16 weeks of age) C57BL/6 mice (SLAC or Shanghai Jihui) were used. Mice were group-housed 4 per cage under a 12-h light–dark cycle (light on from 7:00 to 19:00) with access to food and water ad libitum. All animal studies and experimental procedures were approved by the Animal Care and Use Committee of the animal facility at Zhejiang University.

### CRS

Mice were subjected to CRS stress by placement in 50-ml conical tubes with holes for air flow for 2–4 h per day for 14 consecutive days[10,60].

### Systemic drug delivery for antidepression

All drugs were dissolved in 0.9% saline and injected intraperitoneally. The dose of ketamine (Gutian Pharma or Beikang Pharma) was 10 mg kg$^{-1}$ or 5 mg kg$^{-1}$. The ketamine used in this study was a mixture of $R$-ketamine and $S$-ketamine. At 1 h, 24 h, 3 days and 7 days after drug delivery, animals were subjected to behavioural tests and in vivo recording or killed for in vitro electrophysiology studies.

### Viral vectors

AAV2/9-hSyn-ChrimsonR-tdTomato-WPRE-SV40-pA (titre of $2.27 \times 10^{13}$ viral genomes per ml, 1:10 dilution, 0.1–0.2 µl bilateral into the LH, Taitool) and AAV2/9-hSyn-mCherry-WPRE-pA (titre of $1.53 \times 10^{13}$ viral genomes per ml, 1:10 dilution, 0.1–0.2 µl bilateral into LH, Taitool) were aliquoted and stored at −80 °C until use.

### Surgery

Mice were deeply anaesthetized by 1% sodium pentobarbital (100 mg kg$^{-1}$ body weight, Sigma) and placed in a stereotactic frame (RWD Instruments). The virus was bilaterally injected into the LH (0.1–0.2 µl) (anterior–posterior (AP): −0.82 mm from bregma; medial–lateral (ML), ±1.08 mm; dorsal–ventral (DV), −4.90 mm from the dura) using a pulled glass capillary with a pressure microinjector (Picospritzer III, Parker) at a rate of 0.1 µl min$^{-1}$. The injection needle was withdrawn 10 min after the end of the injection. Optical fibres (200 µm width) were implanted above the LHb (AP, −1.72 mm from bregma; ML, ±1.14 mm; DV, −2.40 mm from the dura) at a 15° angle in the ML direction. After surgery, mice recovered from anaesthesia on a heat pad. Mice were euthanized after all experiments to verify the sites of viral injection and optical fibre implantation. Brain sections were cut at 60 µm thickness (Leica, CM1950) and counterstained with DAPI or Hoechst. Fluorescent image acquisition was performed with an Olympus VS120 virtual microscopy slide scanning system. Only data from mice with correct injection sites were used.

### FST

The FST was used to model behavioural despair as previously described[61]. Mice were individually placed in a cylinder (12 cm diameter, 25 cm height) of water (23–25 °C) and swam for 6 min. The test was performed in normal light conditions (30–35 lux). The water depth was set to prevent animals from touching the bottom with their tails or hind limbs. Animal behaviours were videotaped from the side. The duration of immobility during the 2–6 min test was counted offline by an observer blinded to animal treatment. The duration of immobility was defined as the time when animals remained floating or motionless with only movements necessary for keeping balance in the water.

### SPT

The SPT was used to model anhedonia or the inability to feel pleasure as previously described[62]. Mice were single housed and habituated with two bottles of water for 2 days, followed by two bottles of 2% sucrose for another 2 days. Then after 24 h of water deprivation, animals were exposed to one bottle of 2% sucrose and one bottle of water for 2 h in the dark phase. Bottle positions were switched 1 h after the test started. Total consumption of each fluid was measured, and sucrose preference was defined as the average of sucrose consumption ratios during the first and second hours. The sucrose consumption ratio was calculated by dividing the total consumption of sucrose by the total consumption of both water and sucrose.

### RTPA

Mice were placed in a white open chamber (52 × 26 × 23 cm) consisting of two chambers and allowed to freely move between chambers. During the subsequent 20-min test, a stimulation side was assigned. Laser stimulation (635 nm, 40 Hz, 2-ms pulse, 250 µW) was delivered as soon as mice entered the stimulation side and terminated once mice crossed to the non-stimulation side. A video camera positioned above the chamber recorded each trial. Mouse locations and velocity were tracked and analysed using Any-maze software (Stoelting). The time in the stimulation (light on) and in the non-stimulation (light off) chambers were calculated using Any-maze software.

### Cannula infusion experiment

A double-guide cannula (centre-to-centre distance of 1 mm, RWD) was placed and inserted bilaterally into the LHb (AP, −1.80 mm from bregma; ML, ±0.50 mm; DV, −2.65 mm from the dura) of mice. A double-dummy cannula (RWD), secured with a dust cap, was inserted into the guide cannula to prevent clogging during the recovery period. After mice had recovered for at least 7 days, drugs were microinjected with a double injector cannula while mice were anaesthetized with isoflurane (RWD) on an anaesthetic machine. The extensions were manually sharpened before insertion. Ketamine and memantine (Sigma) were dissolved in 0.9% saline. Before local drug infusion, a tip-sharpened double-injector cannula was inserted into the guide cannula to ensure clear passage and then pulled out. One microlitre of the drug was infused (0.1 µl per 2.5 min) into each side through another set of tip-sharpened double injector cannulae, which were connected to the microsyringe (Baige). The injector cannulae were left in place for an additional 5 min to minimize the spread of the drug along the injection track. Behavioural tests were performed 1 h or 24 h after memantine infusion and 24 h, 7 days or 14 days after ketamine infusion. To verify the sites of drug infusion, 0.2 µl CTB-488 (Invitrogen) was injected to each side of the LHb after all behavioural tests. Mice were then euthanized 30 min after CTB-488 injection for immunostaining. Brain slices were counterstained with Hoechst before mounting onto slides. Fluorescent image acquisition was performed with an Olympus VS120 virtual microscopy slide scanning system. Only data from mice with correctly sited injections were used.

### LC–MS/MS measurement of drug concentration

Male mice were anaesthetized by isoflurane and subsequently decapitated 2 min, 5 min, 10 min, 30 min, 1 h, 4 h and 24 h after intraperitoneal ketamine administration. Blood samples were collected into tubes containing 50 µl EDTA-2Na buffer (30 mg ml$^{-1}$) and centrifuged at 2,000 r.p.m. for 10 min (4 °C), and then the plasma was collected and stored at −80 °C until analysis. Whole brain tissues (about 0.4–0.5 g) were immediately collected into Eppendorf tubes. The tissue samples were immediately frozen in liquid nitrogen and stored at −80 °C until analysis. After samples were thawed, 100 mg brain tissue or 100 µl plasma was measured, and 1 ml solution (ddH$_2$O: acetonitrile = 9:1, v/v) and steel balls were added to homogenize samples for 3 min at 60 Hz with a tissue grinder. A volume of 2.5 µl of ketamine of different concentrations (0.05, 0.1, 0.2, 0.5, 1, 2, 5, 10 and 50 ng ml$^{-1}$) was mixed with 47.5 µl untreated brain tissue to establish a standard calibration curve. The quantification of ketamine was accomplished by calculating area ratios using verapamil solution (2 ng ml$^{-1}$ solution) as the

internal standard. Next, 50 µl of the tested brain tissue was mixed with 200 µl acetonitrile with verapamil solution (2 ng ml$^{-1}$) for vortex mixing. After centrifugation for 15 min at 20,000 r.p.m. at 4 °C, the upper layer was injected into a chromatographic system. The concentrations of ketamine were determined by achiral LC–MS/MS following a previously described method[9,10,63] with slight modifications. The analysis was accomplished using a Waters Acquity UPLC BEH C18 column (2.1 mm × 50 mm inner diameter, 1.7 µm; Waters). The mobile phase consisted of 0.1% formic acid buffer as component A and acetonitrile as component B at a flow rate of 0.35 ml min$^{-1}$, temporized at 10 °C (injection volume: ketamine 10 µl). A linear gradient was run as follows: 0–0.5 min, 10% buffer B; 0.5–1.5 min, from 10% buffer B increased to 90% buffer B; 1.5–2.5 min, 90% buffer B; 2.5–2.51 min, from 90% buffer B decreased to 10% buffer B; 2.51–3.5 min, 10% buffer B. The MS/MS analysis was performed using Waters TQ-S micro. Positive electrospray ionization data were acquired using multiple reaction monitoring using the following transitions for (R,S)-ketamine studies: 238.096 → 124.987.

### Optogenetic light delivery and protocols

For in vitro experiments, LHb brain slices were prepared from mice expressing ChrimsonR[64] in the LH, and then a 635 nm, 40 Hz, 0.5 mW, 2-ms duration pulsed (4 pulses) red light was delivered to activate LH–LHb axon terminals. For the RTPA test, in mice expressing ChrimsonR or mCherry in the LH, a 635 nm red light was bilaterally delivered into the LHb through optical fibres at 40 Hz, 2-ms pulse, 250 µW by laser (Inper) when mice entered the stimulus side of the chamber. For in vivo ketamine binding and unbinding experiments, the same optogenetic protocol (635 nm, 40 Hz, 2-ms pulse, 250 µW) was delivered for 3 min either immediately after ketamine (5 mg kg$^{-1}$) or saline injection or 1 h after ketamine (10 mg kg$^{-1}$) or saline injection.

### Brain slice preparation

Mice were anaesthetized by 1% sodium pentobarbital (100 mg kg$^{-1}$ body weight, Sigma) and then perfused with 20 ml ice-cold ACSF (oxygenated with 95% $O_2$ and 5% $CO_2$) containing (mM): 125 NaCl, 2.5 KCl, 25 NaHCO$_3$, 1.25 NaH$_2$PO$_4$, 1 MgCl$_2$, 2 CaCl$_2$ and 25 glucose, with 1 mM pyruvate added. The brain was removed as quickly as possible after decapitation and put into chilled and oxygenated ACSF. Coronal (for spontaneous neuronal activity recording) or sagittal (for eEPSC recording) slices containing the LHb or the hippocampal CA1 were sliced into 300 µm sections in cold ACSF using a Leica VT1200S vibratome and then transferred to ASCF at 32 °C for incubation and recovery. ACSF was continuously gassed with 95% $O_2$ and 5% $CO_2$. Slices were allowed to recover for at least 1 h before recording. CRS mice and naive mice both went through a FST at least 1 day before brain slice recording, and we only used the CRS animals that showed high durations of immobility (>120 s) and the naive animals that showed low durations of immobility (<120 s) in the FST for slice recording.

### In vitro electrophysiological recordings

For LHb neuron recordings, currents were measured under whole-cell patch-clamp using pipettes with a typical resistance of 4–8 MΩ. For spontaneous neuronal activity recording, the pipettes were filled with internal solution containing (mM): 105 potassium gluconate, 30 KCl, 4 Mg-ATP, 0.3 Na-GTP, 0.3 EGTA, 10 HEPES and 10 sodium phosphocreatine, with pH set to 7.25–7.30. For eEPSC recording, the pipettes were filled with internal solution containing (mM): 115 CsMeSO$_3$, 20 CsCl, 10 HEPES, 2.5 MgCl$_2$, 4 Na-ATP, 0.4 Na-GTP, 10 Na-phosphocreatine, 0.6 EGTA and 5 QX-314, with pH set to 7.25–7.30. The external ACSF solution contained (mM): 125 NaCl, 2.5 KCl, 25 NaHCO$_3$, 1.25 NaH$_2$PO$_4$, 1 MgCl$_2$, 2 CaCl$_2$ and 25 glucose. Cells were visualized with infrared optics on an upright microscope (BX51WI, Olympus). A MultiClamp 700B amplifier controlled by a DigiData 1550 digitizer and pCLAMP10 software were used for electrophysiology recordings (Axon Instruments). The series

resistance and capacitance were automatically compensated after a stable Gigaseal was formed. Recordings were typically performed between 3 and 10 min after break-in.

Spontaneous neuronal activity was recorded under current clamp ($I$ = 0 pA). LHb neurons show three modes of spontaneous activity at resting conditions. Silent cells show no spike activity during recording. Tonic cells spontaneously generate tonic trains of action potentials. Burst cells spontaneously generate clusters of spikes with an initially high but progressively declining intra-burst firing frequency in each burst. After membrane potential stabilized, 3 min of data were collected to calculate bursting spike frequency and bursts per minute. The bursting spike frequency was calculated as the spike number of bursting spikes per second. Bursts per minute was calculated as the number of bursts per minute. The percentage of blockade of bursting spike frequency or bursts per min at each time point were calculated as follows: percentage blockade = (average value of saline group − value of ketamine group)/average value of saline group × 100.

Evoked EPSCs were recorded under voltage clamp at −70 mV or +40 mV in sagittal LHb slices by stimulating the input stria medullaris fibre in a modified extracellular ACSF solution with the GABA$_A$R blocker picrotoxin (100 µM, Tocris). Stimulation pulses (0.25–1.50 mA, 0.2-ms, step by 0.25 mA) were delivered every 6–10 s. Cells were first held at −70 mV to record electrically evoked fast AMPAR-mediated currents (AMPAR-eEPSCs). Subsequently, cells were held at +40 mV to record a combination of AMPAR-mediated and slower NMDAR-mediated currents (NMDAR-eEPSCs)[65,66]. More than three traces were averaged at each stimulation intensity and holding potential. AMPAR-eEPSCs were determined on the basis of the peak current amplitude at −70 mV. NMDAR-eEPSCs were determined on the basis of the current amplitude 35 ms after stimulation onset at +40 mV. NMDA/AMPA ratios were determined by dividing the NMDAR-eEPSCs by the AMPAR-eEPSCs at 1.5 mA stimulation intensity. The percentages of neurons with >10 pA NMDAR-eEPSCs were calculated at 1.5 mA stimulation intensity. The percentage of blockade of NMDAR-eEPSCs at each time point was calculated as follows: percentage blockade = (average value of saline group − value of ketamine group)/average value of saline group × 100.

AMPARs in the LHb mostly lack the GluR2 subunit, and consequently AMPAR-eEPSCs show strong inward rectification and fast decay[11,17], much faster than NMDAR-eEPSCs. To confirm the NMDAR component of the recorded current at 35 ms after stimulation onset, mixed NMDAR-eEPSCs and AMPAR-eEPSCs were recorded at +40 mV with ACSF containing picrotoxin (100 µM, Tocris). Then, 10 µM NBQX (an AMPAR blocker, Sigma) was perfused into the recording solution to block AMPAR currents. Afterwards, 50 µM AP5 (an NMDAR blocker, Sigma) was further added and perfused into the recording solution to block NMDAR currents. The current amplitude detected at 35 ms at +40 mV was significantly blocked by AP5, but showed no difference in the presence or absence of NBQX, which suggested that AMPAR-eEPSCs have mostly decayed by 35 ms (Extended Data Fig. 4a). The pure NMDAR-eEPSCs in LHb neurons 1 h or 24 h after ketamine intraperitoneal injection were recorded at +40 mV using the pharmacological isolation method (with NBQX and picrotoxin in recording ACSF to block AMPARs and GABA$_A$Rs, respectively). NMDAR-eEPSCs were calculated on the basis of the peak amplitude at this recording condition.

For ketamine washout experiments, evoked NMDAR-eEPSCs were recorded under voltage clamp at −70 mV in a modified extracellular ACSF solution with NBQX (10 µM, Sigma) to block AMPARs and with picrotoxin (100 µM, TOCRIS) to block GABA$_A$Rs. Recordings were made in ACSF containing no added Mg$^{2+}$ to reduce the Mg$^{2+}$ blockade of NMDARs. Stimulation intensity (0.1–0.3 ms, 0.1–5 mA) was adjusted for each cell to produce adequate responses. LHb neurons with NMDAR-eEPSCs less than 10 pA were not used in the washout experiments. Stimulation pulses were delivered every 10 s. After 5 min

of stable baseline recording, 100 μM ketamine, 10 μM ketamine or 100 μM memantine was washed into the recording ACSF and then washed out 10 min later to watch for the recovery of NMDAR-eEPSCs in the next 50 min. The vehicle group was carried out as the control. NMDAR-eEPSCs were normalized to the baseline before drug application. The normalized NMDAR-eEPSCs at the end of drug perfusion (at 10 min) were calculated to show the degree of drug blockade (Fig. 3b), and the averaged normalized NMDAR-eEPSCs at 50–60 min were used to show the degree of response recovery (Figs. 3b and 4b). The amount of blockade was calculated in a 10-min bin, and the maximal one was taken as the maximal blockade. Note that owing to the trapping effect, maximal blockade could occur after the end of the 10 min wash-in period (Extended Data Fig. 7). The percentage recovery was calculated as follows: maximal blockade − blockade at 50–60 min (Extended Data Fig. 7). The input resistance during recording was monitored using a 20 mV potential injection.

To prove the efficacy of Dyngo-4a in blocking endocytosis, we tested it on a low-frequency stimulation (LFS)-induced long-term depression protocol in hippocampal CA1 neurons[36]. Sagittal slices containing hippocampal CA1 were prepared, and NMDAR-eEPSCs were recorded under voltage clamp at −70 mV as described above by stimulating the Schaffer collaterals. After 5 min of stable baseline recording, LFS (1 Hz, 15 min, 900 pulses) was performed and then returned to the recording frequency of 0.1 Hz for 35 min. For blockade of NMDAR endocytosis, 30 μM Dyngo-4a (ApexBio) was added to the recording ACSF in LFS-induced long-term depression for the NMDAR-EPSC experiments and washout experiments[67].

For in vitro kick-off experiments, 10 min after ketamine washout, two kick-off sessions (pairing presynaptic 1 Hz electrical stimulation with 3 s postsynaptic depolarization to +10 mV, 3 s pairing per min, 5 min per session, 30 spikes total) with a 4-min interval were performed to cells in the kick-off group. For cells in the pre-alone group, during the same time window (10 min after ketamine washout), two sessions of presynaptic 1 Hz electrical stimulation alone were performed. The average normalized NMDAR-eEPSCs after 50 min were calculated to show the degree of response recovery. As control, two kick-off sessions were applied to cells without ketamine treatment (Extended Data Fig. 8).

## In vivo electrophysiology

For in vivo single-unit recording, a custom-made screw-driven microdrive consisting of an electrode composed of 16 nickel chromium alloy wires (California Fine Wire Company, 0.0014", Stablohm 650) was implanted into the LHb (AP, −1.72 mm; ML, 0.47 mm; DV, −2.50 mm from the dura) of CRS mice. Silver wires were attached to two screws on the skull as ground. The microdrive was secured to the skull with dental cement. After recovery for more than 1 week, mice were allowed to adapt to the recording headstage for 30 min before recording. Spontaneous spiking activity and wideband electronic signals (0.1–7,500 Hz) were recorded using a neural recording system (Plexon) and digitized at 40 kHz with a gain of 1,000×. Spontaneous spiking signals were band-pass-filtered between 300 and 6,000 Hz. Common median reference was assigned as a digital reference. The amplitude threshold for the spike capture was adjusted for each unit according to the signal-to-noise ratio. Spontaneous spiking signals of the mice were recorded for 10 min after habituation in their home cages as the baseline. Spiking signals were continuously recorded for 1 h after ketamine treatment (10 mg kg$^{-1}$, intraperitoneal) with the headstage on the mouse. A paired statistical method was used for the 0–1 h data. For data collected at 24 h or 3 days after ketamine treatment, as the animals were returned to the homecage to rest, the headstage was removed and remounted. Because the headstage was removed and remounted, there was a possibility that the recording electrode had shifted and the number of units had changed. Therefore, we defined the units before and after the headstage reset as different units and used an unpaired

statistical method for the 24 h and 3 days data. The electrodes were lowered in steps of 62.5 μm after each recording session, followed by at least a 3 days of recovery. If mice received a second ketamine injection, at least a 1-week interval was introduced before the next recording session. The CRS animals that showed high immobile duration (>120 s) in the FST were used for in vivo recording. The positions of the electrodes were verified at the end of all experiments, and only data from mice with correct electrode positions were used.

## Spike sorting

All waveforms recorded from each electrode were imported into Offline Sorter V3/4 (Plexon). Single units were manually identified by threshold crossing and principal component analysis. Spikes with an inter-spike interval less than the refractory period (1 ms) were excluded. Cross correlograms were plotted to ensure that no cell was discriminated more than once on the overlapping electrode.

## Data analysis

Data were analysed using Neuroexplorer4/5 (Plexon) and MatLab. We defined in vivo bursting as clusters of spikes beginning with a maximal inter-spike interval of 20 ms and ending with a minimal inter-spike interval of 50 ms. The minimal intra-burst interval was set at 50 ms and the minimal number of spikes in a burst was set at 2. Bursts per minute and bursting spike frequency were analysed using Neuroexplorer4/5 (Plexon) and Excel 2013. For the 1 h continuous recording data after ketamine treatment, the inhibited or excited units (coloured dots in Extended Data Fig. 3) were statistically analysed using $z$-score transformation (bursting spike frequency). The post-injection $z$-score of each unit was calculated as follows:

$$Z = (\bar{x} - \mu)/\sigma$$

where $\bar{x}$ is the mean of all the 100-s-bin values during the post-injection period (that is, 20–30 min after drug injection), $\mu$ and $\sigma$ are the mean and the standard deviation, respectively, of all the 100-s-bin values during the baseline period (that is, 10 min before drug injection).

The inhibition or excitation of a unit was identified when its post-injection $z$-score (bursting spike frequency) was ≤ −1.67 or ≥ 1.67 ($P < 0.05$)[68], respectively. We divided the units into two groups: basal bursting spike frequency >2 Hz group and basal bursting spike frequency <2 Hz group. The percentages of inhibited units in each group were calculated. The percentage of blockade of bursting spike frequency or bursts per min at each time point was calculated as follows: percentage blockade = (baseline value − value of each time point)/baseline value × 100.

## Statistical analysis

Required sample sizes were estimated on the basis of our previous experience performing similar experiments. Mice were randomly assigned to treatment groups. Analyses were performed in a manner blinded to treatment assignments in all behavioural experiments. Statistical analyses were performed using GraphPad Prism software v.7. Based on pre-established criteria, values were excluded from analysis if the drug delivery sites, virus injection sites or electrode implant sites were out of the LHb. All statistical tests were two-tailed, and significance was assigned at $P < 0.05$. Normality and equal variances between group samples were assessed using the D'Agostino and Pearson omnibus normality test and Brown–Forsythe tests, respectively. When normality and equal variance between sample groups were achieved, paired or unpaired $t$-test was used. When normality or equal variance of samples failed, the Mann–Whitney test or Wilcoxon matched pairs test was performed. Linear regression test, Chi-square test, repeated-measures (RM), one-way analysis of variance (ANOVA) or two-way analysis of variance with multiple comparisons were used as appropriate. More details are provided in Supplementary Table 1.

## Reporting summary

Further information on research design is available in the Nature Portfolio Reporting Summary linked to this article.

## Data availability

All data are available in the manuscript or the supplementary materials.

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

**Acknowledgements** We thank Q. Li, G. Pei, L. Wang, C. A. Zarate, W. Hu, W. Chen, C. Zhang, S. Zhu and H. Wu for stimulating discussions; Y. Ping for help with LC–MS/MS; W. Zhou of the Core Facilities, Zhejiang University Medical Center/Liangzhu laboratory for technical support; and Q. Xin, Z. Ni and colleagues from the Hu Laboratory for assistance in experiments. This work was supported by the STI2030-Major Projects (2021ZD0203000 (2021ZD0203001)), the National Natural Science Foundation of China (31830032, 32130042 and 82288101), the Key-Area Research and Development Program of Guangdong Province (2018B030334001 and 2018B030331001), the Leading Innovation and Entrepreneurship Team in Zhejiang Province (2020R01001), the Starry Night Science Fund of Zhejiang University Shanghai Institute for Advanced Study (SN-ZJU-SIAS-002), the New Cornerstone Science Foundation, the Project for Hangzhou Medical Disciplines of Excellence, and Key Project for Hangzhou Medical Disciplines to H.H. C.J.L. is supported in part by R35GM118114 from NIGMS.

**Author contributions** S.M., M.C. and H.H. designed the study and analysed the data. S.M. conducted the behavioural pharmacology experiments. M.C. performed the in vitro patch-clamp experiments with the assistance of Z.W., S.L., Y.J., J.W. and Y.C. M.C. and X.X. performed the in vivo recordings and in vivo data analyses. S.M., S.W. and M.C. performed the optogenetic experiments. H.H. conceived the project and wrote the manuscript with the assistance of M.C. and S.M. C.J.L., Y.Y., H.L., S.D., H.M., Y. Zhang and Y. Zhu contributed to experimental design and manuscript writing.

**Competing interests** The authors declare no competing interests.

**Additional information**
**Correspondence and requests for materials** should be addressed to Hailan Hu.

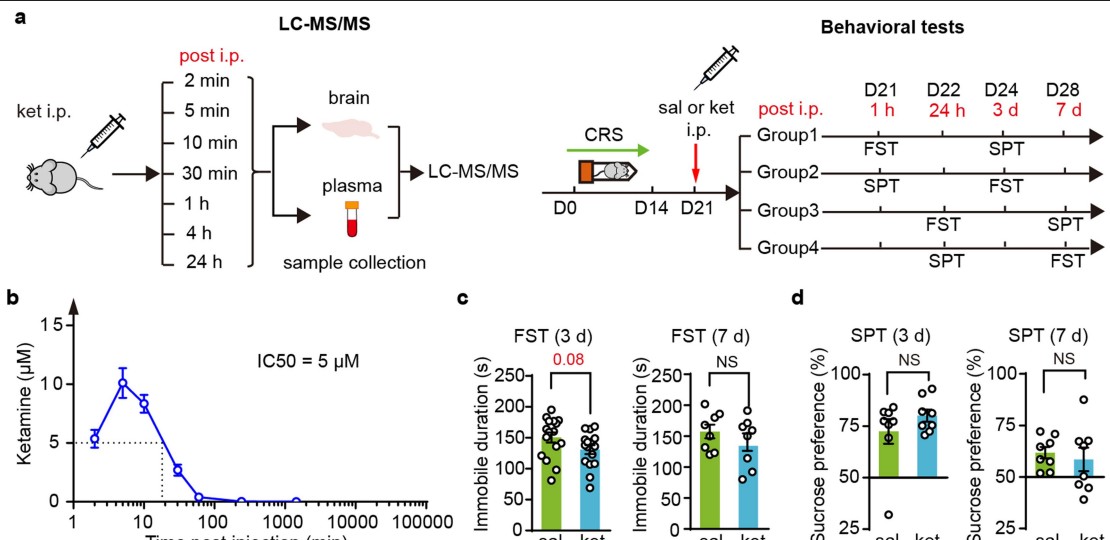

**Extended Data Fig. 1 | Single injection of ketamine no longer causes antidepressant effects on day 3 or day 7. a**, Schematic of LC-MS/MS (left) and experimental paradigm for behavioral tests (right) after i.p. injection of 10 mg kg⁻¹ ketamine. **b**, Plasma concentration of ketamine after a single i.p injection of ketamine in CRS mice, as measured by LC-MS/MS. Dotted line indicates the half-life of ketamine. **c**, **d**, Behavioral effects at 3 d and 7 d after a single i.p. injection of ketamine in CRS mice in the FST (**c**) and the SPT (**d**). Sal: saline. Ket: ketamine. NS, not significant. Error bars indicate SEMs. (see Supplementary Table 1 for statistical analyses and *n* numbers). This figure is related to Fig. 1.

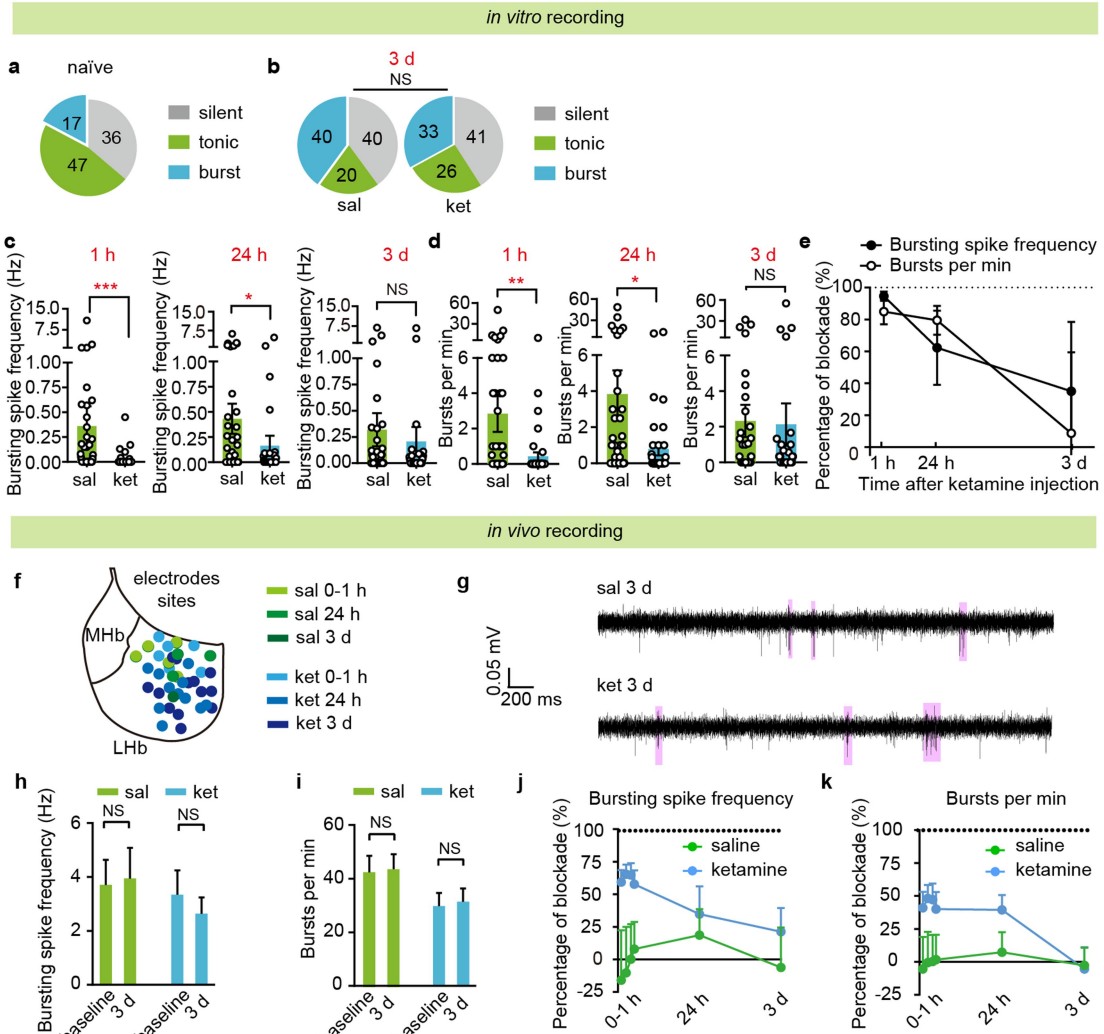

**Extended Data Fig. 2 | Single injection of ketamine causes prolonged suppression of LHb bursting activity. a**, Pie charts illustrating the percent abundance of the three types of LHb neurons in naïve mice. **b**, Pie charts illustrating the percent abundance of the three types of LHb neurons in CRS mice 3 d after saline or ketamine i.p. injection. **c**, **d**, Bar graphs illustrating the bursting spike frequency (number of bursting spikes per second,**c**) and bursts per min (number of bursts per minute, **d**) in CRS mice at different time points after saline or ketamine i.p. injection. **e**, Percentage of blockade of bursting spike frequency or bursts per min at each time point calculated as (saline value - ketamine value)/saline value. **f**, Recording sites of electrodes in LHb. Black lines indicate location of habenula, green dots indicate recording sites of saline

group, blue dots indicate recording sites of ketamine group. **g**, Example traces showing in vivo neuronal activity recorded in CRS mice 3 d after saline or ketamine administration. Bursts (pink shades) are identified by the ISI method (see Methods). **h**, **i**, Bar graphs illustrating the bursting spike frequency (**h**) and bursts per minute (**i**) in CRS mice 3 d after saline or ketamine i.p. injection. **j**, **k**, Percentage of blockade of bursting spike frequency (**j**) and bursts per minute (**k**) at each time point calculated as (baseline value - value of each time point)/baseline value. * $P < 0.05$; ** $P < 0.01$; *** $P < 0.001$; NS, not significant. Error bars indicate SEMs. (see Supplementary Table 1 for statistical analyses and $n$ numbers). This figure is related to Fig. 1.

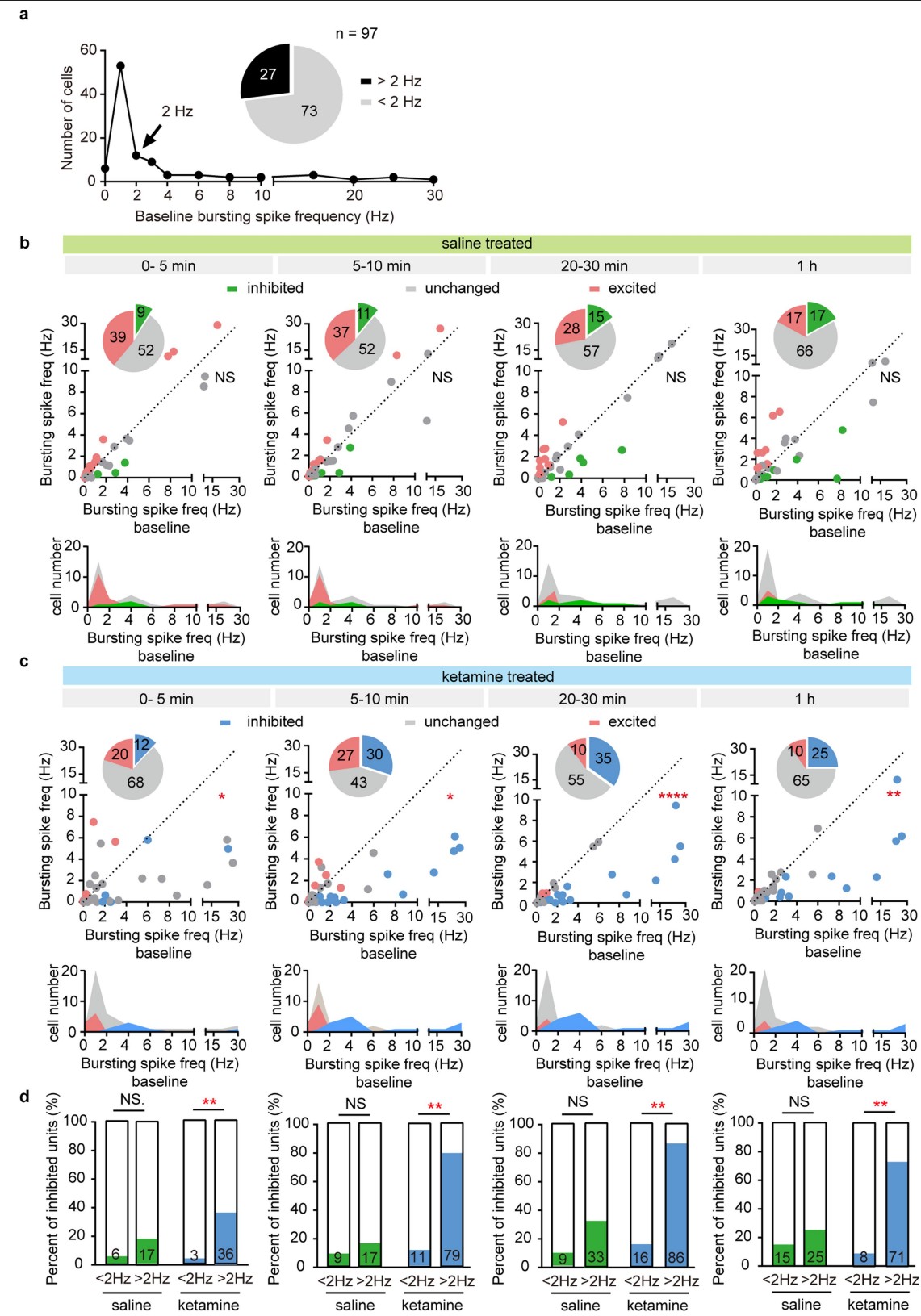

**Extended Data Fig. 3** | See next page for caption.

**Extended Data Fig. 3 | Ketamine preferentially inhibits LHb neurons with high bursting spike frequency. a**, Histogram distribution of baseline bursting spike frequency of all recorded LHb units. Pie graph showing percentage of neurons with baseline bursting spike frequency larger than 2 Hz (black) and smaller than 2 Hz (grey). **b**, **c**, Top: Scatter plots of the bursting spike frequency of recorded LHb units at baseline state plotted against bursting spike frequency at 0–5 min, 5–10 min, 20–30 min and 1 h after i.p. injection of saline (**b**) or ketamine (**c**). Green/blue, grey and red dots indicate neurons showing significant inhibition, no change and significant increase in bursting spike frequency, respectively. Pie graphs show the percentage of inhibited (green in saline group, **b**; blue in ketamine group, **c**), excited (red) and unchanged (grey) LHb neurons. Bottom: histogram distribution of baseline bursting spike frequency of the three LHb neuron types. **d**, Percentage of LHb neurons inhibited by either saline (green) or ketamine (blue) in the "<2 Hz" and ">2 Hz" (baseline bursting spike frequency) groups. Numbers in each box are percentages of inhibited neurons. Note that a significantly larger fraction of neurons is inhibited by ketamine in the ">2 Hz group". *$P < 0.05$; **$P < 0.01$; ****$P < 0.0001$; NS, not significant. (see Supplementary Table 1 for statistical analyses and $n$ numbers). This figure is related to Fig. 1.

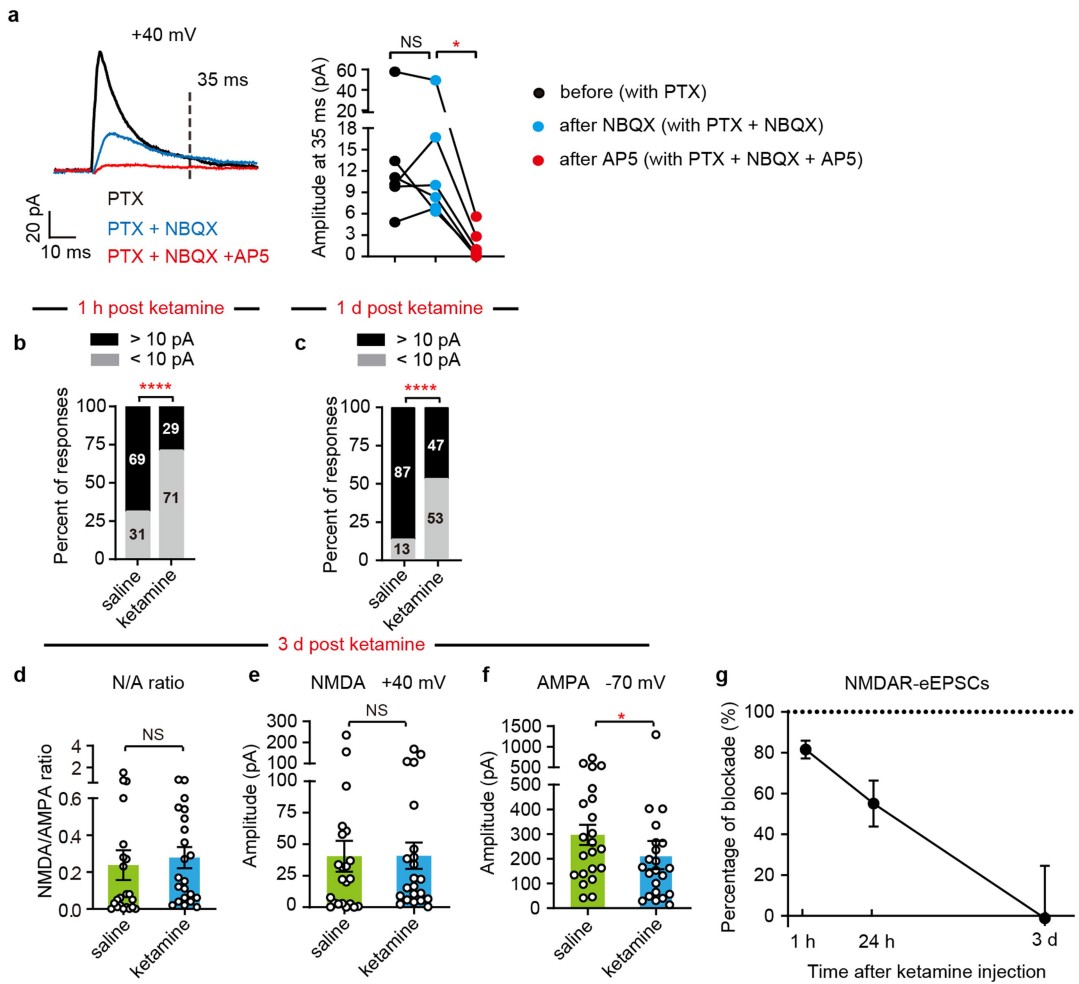

**Extended Data Fig. 4 | Single injection of ketamine in CRS mice causes prolonged inhibition of NMDAR currents in LHb. a**, Left: example traces of LHb eEPSCs evoked at +40 mV in presence of PTX (black), PTX + NBQX (blue) and PTX + NBQX + AP5 (red) in ACSF. Right: Amplitudes of eEPSCs measured at 35 ms after stimulation onset. Note that eEPSC amplitudes at 35 ms post stimulation are not significantly affected by NBQX blockade of AMPARs, but are greatly reduced by AP5 blockade of NMDARs, indicating that the major component at this time point is NMDAR-eEPSC. **b, c**, Bar graphs showing percentage of NMDAR-eEPSCs smaller or larger than 10 pA at 1 h (**b**) and 24 h (**c**) after saline or ketamine treatment. **d-f**, Bar graphs of ratios of NMDAR-eEPSCs and AMPAR-eEPSCs (**d**), NMDAR-eEPSCs (**e**) and AMPAR-eEPSCs (**f**) of LHb neurons at 1.5 mA stimulation intensity from brain slices prepared at 3 d after i.p. injection of saline or ketamine in CRS mice. **g**, Percentage of blockade of NMDAR-eEPSCs at each time point calculated as (saline value - ketamine value)/ saline value. * $P < 0.05$; **** $P < 0.0001$; NS, not significant. Error bars indicate SEMs. (see Supplementary Table 1 for statistical analyses and $n$ numbers). This figure is related to Fig. 2.

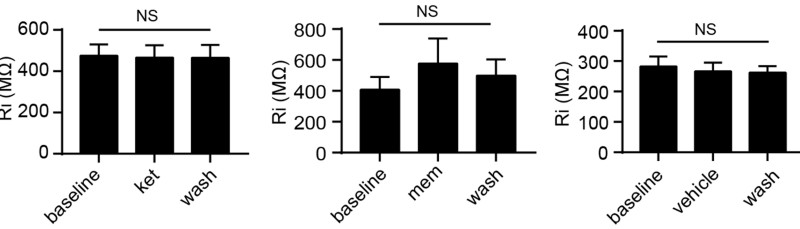

**Extended Data Fig. 5 | Input resistance of recorded neurons.** Input resistance of recorded neurons during baseline, wash-in (at 10 min) and wash-out (at 50 min) periods in ket-, mem- and vehicle- treated groups. NS, not significant. Error bars indicate SEMs. (see Supplementary Table 1 for statistical analyses and $n$ numbers). This figure is related to Fig. 3a.

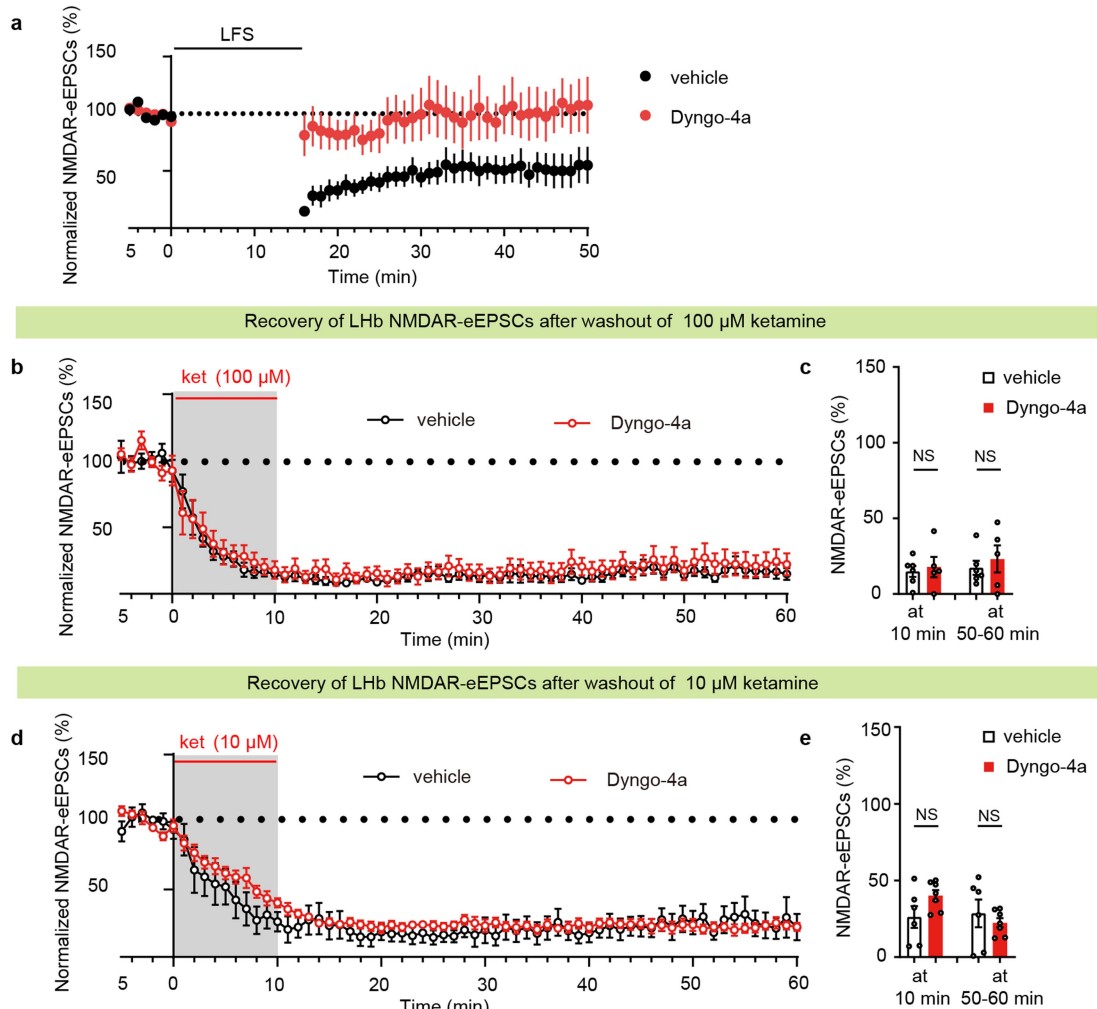

**Extended Data Fig. 6 | Recovery of LHb NMDAR-eEPSCs after washout of ketamine in presence of endocytosis blocker Dyngo-4a. a**, Dyngo-4a successfully blocks LFS (1 Hz, 15 min)-induced, endocytosis-based long-term depression of NMDAR-eEPSCs in hippocampal CA1 neurons. NMDAR-eEPSCs are isolated by application of PTX and NBQX in $Mg^{2+}$ free ACSF under voltage clamp at −70 mV. In the Dyngo-4a group, 30 μM Dyngo-4a is additionally added in ACSF throughout the recording. **b**, **d**, In presence of Dyngo-4a, LHb NMDAR-eEPSCs (normalized by baseline) still do not show recovery after wash-out of 100 μM ketamine (**b**) or 10 μM ketamine (**d**) in LHb neurons. **c**, **e**, Bar graphs showing NMDAR-eEPSCs at the end of the 10 min perfusion period (left) and at 50–60 min (right). Ket: ketamine. The data of vehicle group in (**b**) was from the data of ket- group in Fig. 3a and the data of vehicle group in (**d**) was from the data of "no stimulus" group in Fig. 4a. NS, not significant. Error bars indicate SEMs. (see Supplementary Table 1 for statistical analyses and $n$ numbers).

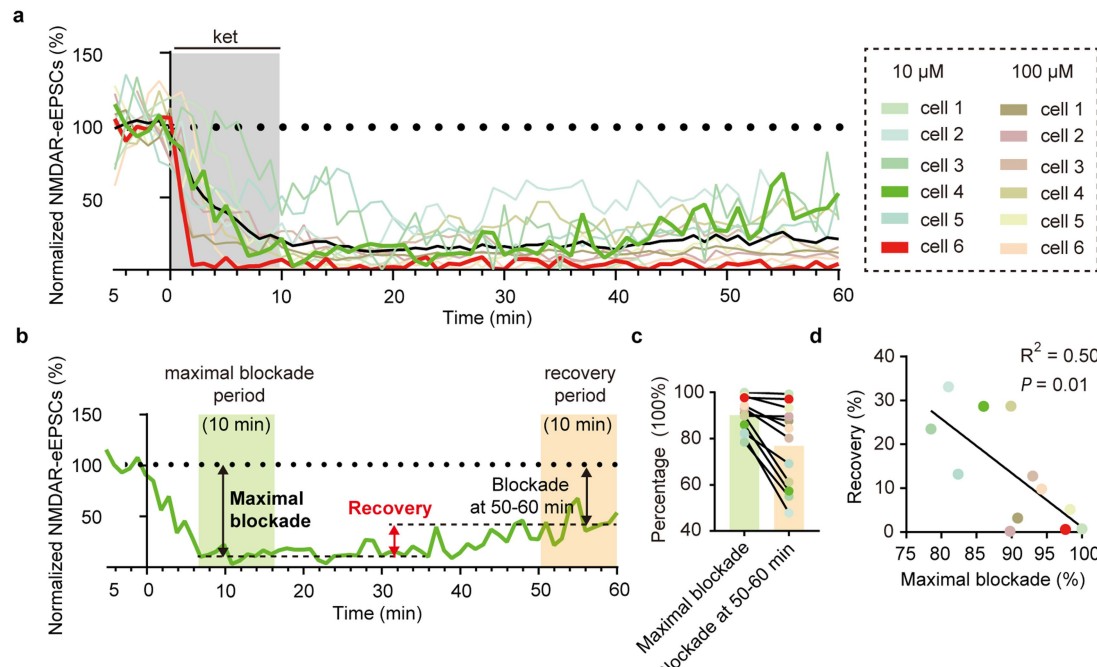

**Extended Data Fig. 7 | Potential reverse correlation between the level of initial NMDAR blockade and the level of recovery. a**, NMDAR-eEPSCs (normalized by baseline) during incubation and wash-out of ketamine (10 µM or 100 µM). NMDAR-eEPSCs are isolated by application of PTX and NBQX in $Mg^{2+}$ free ACSF under voltage clamp at −70 mV. Bin is 1 min. Each line represents one recorded cell. The red line shows an example cell with no recovery and the green line shows an example cell with 29% recovery. The black line is the average of all recorded cells. **b**, Illustration of the calculation of maximal blockade, blockade at 50–60 min and recovery percentage. The maximal blockade is averaged by the 10-min values during maximal blockade period (there is some variation among individuals, but mostly between 0 min to 20 min). Recovery percentage is calculated as: maximal blockade – blockade at 50–60 min. **c**, Bar graph showing the maximal blockade and blockade at 50–60 min of each cell after ketamine perfusion. Each color represents one recorded cell. **d**, Recovery percentage plots against maximal blockade of LHb-NMDAR-eEPSCs. Each color represents one recorded cell. (see Supplementary Table 1 for statistical analyses and *n* numbers).

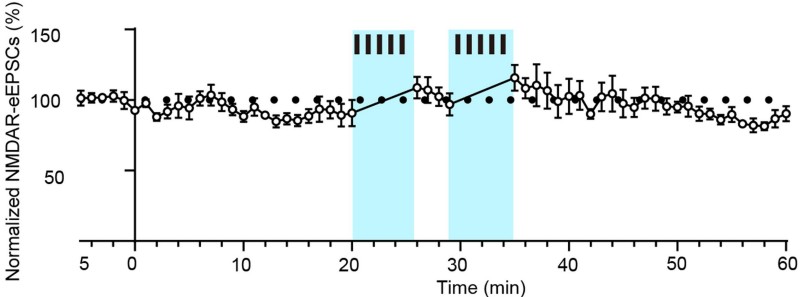

**Extended Data Fig. 8 | "Kick off" protocol does not induce NMDAR-LTP.**
NMDAR-eEPSCs (normalized by baseline) after kick off protocol. NMDAR-eEPSCs are isolated by application of PTX and NBQX in $Mg^{2+}$ free ACSF under voltage clamp at −70 mV. Error bars indicate SEMs. (see Supplementary Table 1 for statistical analyses and $n$ numbers).

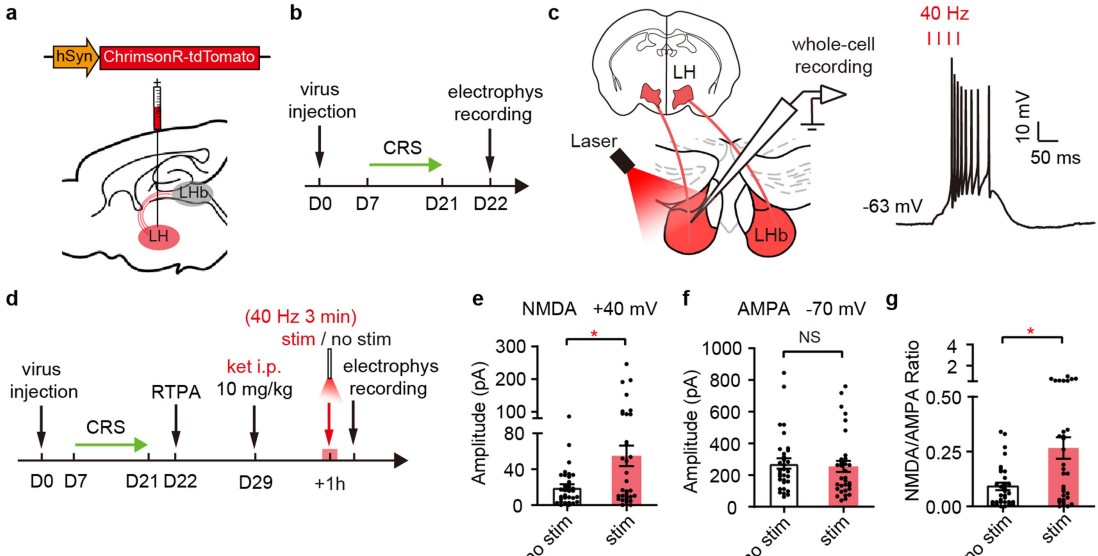

**Extended Data Fig. 9 | 40 Hz LH-LHb stimulation induces LHb burst firing and kicks off trapped ketamine in vivo. a**, Schematic of viral construct, viral injection in the LH of CRS mice. **b**, Experimental paradigm. **c**, Schematic of whole-cell recording of LHb neurons in vitro and representative trace showing burst firing elicited by pulsed light stimulation (635 nm, 40 Hz, 2 ms pulse, 0.5 mW) of LH terminals in LHb brain slices. **d**, Experimental paradigm for electrophysiological recording immediately after optical stimulation in CRS mice. Optical stimulation (635 nm, 40 Hz, 2 ms pulse, 3 min) of LH-LHb terminals or no stimulation is delivered at 1 h after ketamine injection. **e-g**, NMDAR-eEPSCs (**e**), AMPAR-eEPSCs (**f**) and NMDA/AMPA ratio (**g**) of LHb neurons recorded immediately after optical stimulation from experiment in (**d**). * *P* < 0.05, NS, not significant. Error bars indicate SEMs. (see Supplementary Table 1 for statistical analyses and *n* numbers).

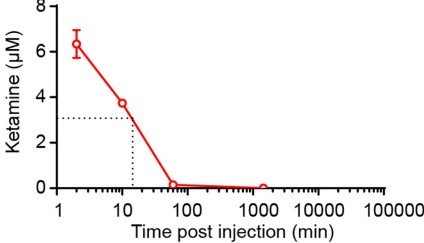

**Extended Data Fig. 10 | Brain concentration of ketamine after a single i.p. injection of 5 mg kg⁻¹ ketamine in CRS mice, as measured by LC-MS/MS.** As noted by the dotted line, the half-life of ketamine is 13 min. Error bars indicate SEMs. (see Supplementary Table 1 for statistical analyses and *n* numbers).

# Reporting Summary

## Statistics

For all statistical analyses, confirm that the following items are present in the figure legend, table legend, main text, or Methods section.

| n/a | Confirmed | |
|---|---|---|
| ☐ | ☒ | The exact sample size ($n$) for each experimental group/condition, given as a discrete number and unit of measurement |
| ☐ | ☒ | A statement on whether measurements were taken from distinct samples or whether the same sample was measured repeatedly |
| ☐ | ☒ | The statistical test(s) used AND whether they are one- or two-sided<br>*Only common tests should be described solely by name; describe more complex techniques in the Methods section.* |
| ☒ | ☐ | A description of all covariates tested |
| ☐ | ☒ | A description of any assumptions or corrections, such as tests of normality and adjustment for multiple comparisons |
| ☐ | ☒ | A full description of the statistical parameters including central tendency (e.g. means) or other basic estimates (e.g. regression coefficient) AND variation (e.g. standard deviation) or associated estimates of uncertainty (e.g. confidence intervals) |
| ☐ | ☒ | For null hypothesis testing, the test statistic (e.g. $F$, $t$, $r$) with confidence intervals, effect sizes, degrees of freedom and $P$ value noted<br>*Give P values as exact values whenever suitable.* |
| ☒ | ☐ | For Bayesian analysis, information on the choice of priors and Markov chain Monte Carlo settings |
| ☒ | ☐ | For hierarchical and complex designs, identification of the appropriate level for tests and full reporting of outcomes |
| ☒ | ☐ | Estimates of effect sizes (e.g. Cohen's $d$, Pearson's $r$), indicating how they were calculated |

*Our web collection on statistics for biologists contains articles on many of the points above.*

## Software and code

Policy information about availability of computer code

| Data collection | N/A |
|---|---|
| Data analysis | GraphPad Prism software v7 was used for statistical analyses.Offline sorter V3 was used for single unit sorting. Neuroexplorer V4 and Matlab was used for analysis of burst and synchronization. |

For manuscripts utilizing custom algorithms or software that are central to the research but not yet described in published literature, software must be made available to editors and reviewers. We strongly encourage code deposition in a community repository (e.g. GitHub). See the Nature Portfolio guidelines for submitting code & software for further information.

## Data

Policy information about availability of data

All manuscripts must include a data availability statement. This statement should provide the following information, where applicable:
- Accession codes, unique identifiers, or web links for publicly available datasets
- A description of any restrictions on data availability
- For clinical datasets or third party data, please ensure that the statement adheres to our policy

The data and primary codes that support the findings of this study data are available from the corresponding author upon reasonable request.

## Human research participants

Policy information about studies involving human research participants and Sex and Gender in Research.

| | |
|---|---|
| Reporting on sex and gender | N/A |
| Population characteristics | N/A |
| Recruitment | N/A |
| Ethics oversight | N/A |

Note that full information on the approval of the study protocol must also be provided in the manuscript.

## Field-specific reporting

Please select the one below that is the best fit for your research. If you are not sure, read the appropriate sections before making your selection.

☒ Life sciences ☐ Behavioural & social sciences ☐ Ecological, evolutionary & environmental sciences

For a reference copy of the document with all sections, see nature.com/documents/nr-reporting-summary-flat.pdf

## Life sciences study design

All studies must disclose on these points even when the disclosure is negative.

| | |
|---|---|
| Sample size | Required sample sizes were estimated based on our past experience performing similar experiments. |
| Data exclusions | Values were excluded from the analyses if the viral injection or drug delivering sites were out of LHb. |
| Replication | The experiment for each experiment was successfully repeated for at least two times. |
| Randomization | Animals were randomly assigned to treatment groups. |
| Blinding | Analysis were performed in a manner blinded to treatment assignments in all behavioral experiments. |

## Reporting for specific materials, systems and methods

We require information from authors about some types of materials, experimental systems and methods used in many studies. Here, indicate whether each material, system or method listed is relevant to your study. If you are not sure if a list item applies to your research, read the appropriate section before selecting a response.

### Materials & experimental systems

| n/a | Involved in the study |
|---|---|
| ☒ | ☐ Antibodies |
| ☒ | ☐ Eukaryotic cell lines |
| ☒ | ☐ Palaeontology and archaeology |
| ☐ | ☒ Animals and other organisms |
| ☒ | ☐ Clinical data |
| ☒ | ☐ Dual use research of concern |

### Methods

| n/a | Involved in the study |
|---|---|
| ☒ | ☐ ChIP-seq |
| ☒ | ☐ Flow cytometry |
| ☒ | ☐ MRI-based neuroimaging |

## Animals and other research organisms

Policy information about studies involving animals; ARRIVE guidelines recommended for reporting animal research, and Sex and Gender in Research

| | |
|---|---|
| Laboratory animals | Male adult (8–16 weeks of age) C57BL/6 mice (SLAC or Shanghai Jihui) were used. Male adult (8-16 weeks of age) C57BL/6 mice (SLACor Shanghai Jihui) were used for establishing the chronic restraint stress (CRS) depression model. |
| Wild animals | No wild animals were used in the study. |

| Reporting on sex | *Indicate if findings apply to only one sex; describe whether sex was considered in study design, methods used for assigning sex. Provide data disaggregated for sex where this information has been collected in the source data as appropriate; provide overall numbers in this Reporting Summary. Please state if this information has not been collected. Report sex-based analyses where performed, justify reasons for lack of sex-based analysis.* |
| --- | --- |
| Field-collected samples | No field collected samples were used in the study. |
| Ethics oversight | All animal studies were approved by the Animal Care and Use Committee of the animal facility at Zhejiang University. |

Note that full information on the approval of the study protocol must also be provided in the manuscript.

