## [Peer Review File · Nature]

Manuscript Title: Sustained Antidepressant Effect of Ketamine via NMDAR Trapping in LHb

Editorial Notes:

Redactions – unpublished data

Reviewer Comments & Author Rebuttals

Reviewer Reports on the Initial Version:

Referees' comments:

Referee #1 (Remarks to the Author):

In light of the fast-acting mode, the use of ketamine for depression treatment has given incredible hope to physicians and patients affected by this disorder. Following the idea to understand how mechanistically ketamine produces its fast yet relatively lengthy actions the group of Hailan Hu in this work carried out experiments employing physiology (in vitro and in vivo) and circuit-based manipulations. The work focuses on the lateral habenula identified previously by the authors to be an anatomical target of ketamine to exert its antidepressant effects.

Following a set of results previously published by the same group, the authors further confirm that a single ketamine injection leads to antidepressant effects, leads to a reduced burst firing of habenular neurons and concomitantly reduces the currents evoked by NMDA receptors activation.

The authors propose that the long-lasting effects produced by ketamine rely on a use-dependent trapping of the compound by the NMDA receptor. According to the authors, such a trapping within the ionotropic receptors can be unlocked by neuronal activity. The use of electrical stimulation and optogenetic activation of hypothalamic afferents restores NMDA currents, which in turns has behavioural repercussions. The authors propose that resolving this mechanism will provide ground to leverage on Ketamine-NMDA receptor interaction to modulate the efficacy of depression treatments.

This work follows closely previous publications from the authors. The same group has shown in an outstanding work that ketamine exerts its effect 1h after exposure via affecting burst firing and NMDA receptor function in habenula neurons (Yang et al., 2018; see also Cui et al., 2019). The authors here increment our knowledge on the understanding how ketamine not only is fast acting but also long-lasting. The study is interesting yet only provides an extension from previous work and a very descriptive sets of results which both reduce the general enthusiasm. Several controls are missing, especially within the electrophysiology experiments. There is a general lack of a solid set of

experiments that demonstrates that Ketamine is actually trapped in the receptors. Structural experiments, single channel experiments and subunit-specific assessment are few of the very important experiments that are missing but necessary to support the conclusions and statements in this work.

Major points

1- The authors employ a model of depression, the CRS, which they have used throughout their previous papers (Yang et al., 2018). The depressive-like phenotype varies across models (stronger anhedonia or behavioural despair depending on the model). Indeed, the authors observe quite of a large variance in the antidepressant efficacy at both 1h and 24h in both FST and SPT. This leads to rather weak statistical effects. Do the authors have any relationship appearing with decay of Ketamine and behaviour? Such a bulk analysis not taking into account individual variability remains outdated compared to the behavioural assertions available with new neurotechnologies (Cerniauskas et al., 2018). The authors should provide evidence for similar timelines of action in another model of depression, which if possible can have better face validity with the human condition.

Minor point. The authors miss some valuable controls for their behavioural assessment. What happens with treatment non-stress animals?

2- The analysis of the burst is interesting however only expands previous findings. In structures including thalamus or VTA the burst activity relies on the RMP of neurons. The example shown indicate a 10mV difference in RMP. Is it the case that systematically cells bursting are more hyperpolarized? A better biophysical report of parameters should be provided.

When reporting the signatures of burst firing the authors miss to provide the control group. Is the CRS leading to increase bursting. This remains unknown and all data seem to relate only to the ketamine effect on the saline in CRS. Such a control is also missing from the in vivo data.

Minor point. The burst signatures provided in the methods differ from previous work published by the authors, thus this makes difficult to compare across literature. Is there a reason to define it in a different manner?

3- The authors have previously shown that burst are regulated by a complex machine that integrates NMDA receptors, calcium channels and even astrocytes. The authors should show specificity for the effect of Ketamine on NMDA and understand whether the other components are similarly affected.

The measurement of NMDA and AMPA receptors currents and their analysis of I/O curves of absolute value is likely not appropriate. In many instances the currents seem also in the nA range which is at the mercy of space clamp issues. There is no doubt that ketamine can reduce NMDA currents as shown in a variety of studies throughout the brain, however the authors should consolidate these findings in order to eliminate the potential issues of stimulus intensity and location which is difficult to reproduce across recordings. This is particularly the case in a structure that has not an organized anatomical architecture. More experiments, in better controlled conditions should be provided.

Majorly the NMDA component is assessed in this work. Yet, the authors have shown that hypothalamic stimulation strongly affects the AMPA component. Furthermore, the group of R. Malinow has shown the GABA component being affected in a model of depression. The authors need to provide solid evidence for the specificity of their effects.

4- The trapping hypothesis proposed by the authors is very intriguing. However, the data fall short to causally demonstrate that ketamine is really trapped within the channel. Furthermore, the trapping idea, as also indicated in their text was previously proposed in other work. This questions the degree of novelty of the study.

What is the rationale in these experiments to use the 100uM dose in Figure 3. The authors nicely show in the initial figures that the real detection of Ketamine was in the order of 16uM. This should be the concentration employed for these experiments as also done in Figure 4. Control experiments monitoring AMPA and GABA_A receptors currents are appropriate in this case. An appropriate control showing that Dyno4a is an efficient blocker of endocytosis of NMDA should be shown as the author cite work using Dynosore rather and in a recombinant system.

The authors claim in Figure 4: These results provide a plausible mechanism for the prolonged blockade of NMDARs after ketamine clearance in plasma. Here, the authors fail to explain this statement. The demonstration that endocytosis is not part of the mechanism is poor (controls missing), there is no alternative strategy tested and the trapping hypothesis remains simply a preference. The authors cite a paper that does not support the trapping hypothesis but rather promote the idea that lateral trafficking contribute to the maintenance of NMDA receptors pool. Altogether this renders this work quite preliminary, and descriptive likely appropriate to more specialized audience.

5- The kick off protocol is intriguing. Also in this case a major control is missing. If the slice is not bathed in ketamine but in saline, and the kick off protocol is provided, is the NMDA component affected. There are instances of plasticity of NMDA in the hippocampus, the VTA and the habenula. This is a major drawback for the claims made by the authors. Is it possible that only postsynaptic activity represents a kick off state? If so, one would think that the conditions to kick off ketamine are larger in a depression state as the activity is higher. This should be tested experimentally.

Why moving to a complete different protocol using higher stimulation in vivo? These presynaptic manipulations can engage plasticity mechanisms at many sites, and this was not controlled for. What the effect of this stimulation with Chrimson at synapses remains completely unknown.

6- The choice and rationale for choosing the hypothalamic input is not clear. The authors published recently the importance of this pathway in stress driven depression. This once again reduces the novelty of the presented work. It would have been instead useful to provide information on the efficiency of multiple inputs onto habenula. The LH is only one among many releasing glutamate. Is the hypothesis valid for every afferent that is excitatory?

Minor point. The optogenetic Chrimson-based experiment lacks some controls. What happens if the

authors swap the concentrations in vivo, and run the same timed stimulation protocols? It remains unclear why the model works not knowing the stoichiometry of NMDA receptors at every different stage of ketamine presence. Is the Kon-Koff model proposed by the authors better to be tested in heterologous systems enabling the choice of specific subunits. This would provide better information as this basic knowledge is missing in this neuronal circuit.

Minor point

- In the introduction the authors state experimental demonstration for the repercussions of habenular bursting on downstream dopamine and serotonin systems. This should be toned down as for the moment it remains a non-tested hypothesis.

- The discussion would need likely some revision. The authors explain their view on how to leverage the trapping mechanisms in the context of therapeutics. This remains far stretched as it is not clear what is the vision of providing negative stimuli to patients. The authors previously indicate the importance of plasticity in the context of stress – now it seems that stress can be leveraged without consequences. This may lead to confusion rather than clarity in the field.

Referee #2 (Remarks to the Author):

Ma et al.

A. Summary of key results: Ketamine is highly effective for treatment of depression, yet its mechanism of action is not understood: especially the paradox that the behavioral effects of ketamine last about 100 times longer than its half-life for elimination. This study presents the extraordinary finding that the prolonged action of ketamine beyond its tissue concentration lifetime is because of prolonged channel block of the NMDAR receptor by ketamine.

B. Originality and significance: To my knowledge the findings are original and highly significant. Now that the mechanism of ketamine is known, as shown in this study to be prolonged channel block, then it should be possible to devise even more effective prolonged blocking agents. The biophysical observations that support the proposed mechanism of channel block are backed up with appropriate behavioral studies to support the conclusion of prolonged block. This paper should be of wide interest in medicine, neuroscience, and biophysics.

C. Data and Methodology: The validity of the approach and the quality of the data support the conclusions drawn.

D. Statistics: Statistics are appropriate.

E. Conclusions: The conclusions are well supported by the presented data.

F. Improvements: I have no suggested improvements or experiments.

G. References: References have been made to previous work, but because I do not work directly in the fields of depression and ketamine, I would not know if all appropriate references have been made.

H. The paper is very well written and the title, abstract, introduction, results, and discussion are clear and appropriate for the conclusions.

No major suggestions.

Minor suggestions are below where the numbers indicate line numbers.

1. Title: If there is room, spell out LHB

68. or the ketamine metabolite

99. because the apparent off rate of ketamine is so slow, it seems that 0.23 μM might still contribute to binding. Comment.

107. There is no e and f in Fig. 1

149. fig. 4 = Fig. 4

160. fig. = Fig

Extended Fig. 8. Rather than repeat the text after the title, perhaps the title should be a conclusion. i.e. Ketamine concentration in the brain falls exponentially with a half time of 12??? minutes or what ever it is.

280. folds should be fold

288 An brief explanation of mCherry and ChrimsonR expressing mice would be useful here to interpret the findings.

331 suggest: and illustrate a case where a biophysical channel block mechanism extended to extraordinarily long times by the in vivo physiological parameters explains an import therapeutic function. (I suspect you can write a better version.)

370 fig. = Fig

924. Laser stimulation (put some brief details here of what laser stimulation was).

Author Rebuttals to Initial Comments:

Referees' comments:

Referee #1:

This work follows closely previous publications from the authors. The same group has shown in an outstanding work that ketamine exerts its effect 1h after exposure via affecting burst firing and NMDA receptor function in habenula neurons (Yang et al., 2018; see also Cui et al., 2019). The authors here increment our knowledge on the understanding how ketamine not only is fast acting but also long-lasting. The study is interesting yet only provides an extension from previous work and a very descriptive sets of results which both reduce the general enthusiasm. Several controls are missing, especially within the electrophysiology experiments. There is a general lack of a solid set of experiments that demonstrates that Ketamine is actually trapped in the receptors. Structural experiments, single channel experiments and subunit-specific assessment are few of the very important experiments that are missing but necessary to support the conclusions and statements in this work.

We appreciate reviewer's positive comment on our previous work. However, we respectfully but strongly disagree with his/her assessment on the novelty of the current manuscript. Below we highlight the novelty of the current findings.

While much progress has been made in understanding ketamine's rapid action, however, much less is understood about mechanisms underlying its sustained effects. Despite a half-life of 13 minutes, ketamine's antidepressant effects in mice can last for at least 24 h. This huge discrepancy poses an interesting basic biological question, and has strong clinical implications. Previously, ketamine's sustained effect was attributed to the *de novo* spine growth, or the ketamine metabolite HNK. However, ketamine-induced spine growth cannot be detected until 12 hours post treatment, and the half-life of HNK is still less than 30 minutes. In the current work, we reveal a much simpler and more direct mechanism: the trapping blockade of NMDARs. We demonstrate that ketamine continues to block NMDARs in the lateral habenula (LHb), for up to 24 h, long after ketamine's plasma elimination. We further demonstrate that this surprisingly long-lasting molecular inhibition is not due to endocytosis, but governed by the unusually long off-rate of the ketamine-NMDAR interaction. Furthermore, ketamine-suppressed NMDAR currents can be restored *in vitro* and *in vivo* upon neural activation and forced NMDAR channel opening. Harnessing these unique biophysical characteristics, we showed that activating the LHb at different ambient ketamine levels ($[K_{out}]$) is able to either shorten or extend ketamine's sustained antidepressant effects *in vivo*. These findings establish a key causal role of the long-term blockade of LHb-NMDARs in mediating ketamine's sustained antidepressant effects, and illustrate a case where

a simple molecular biophysical mechanism explains an important therapeutic function.

In addition, our data also resolves a big puzzle in the field as to why other NMDAR inhibitors, such as memantine, have much less optimal antidepressant effects in the clinics. By comparing the trapping property of ketamine with memantine, our study demonstrates that the unique pharmacodynamics property of ketamine is critical in its antidepressant effects, and that optimization of such property is a promising new direction for developing new antidepressant treatment. As pointed out by Reviewer 2, such study should be of broad interest to medicine, neuroscience, and biophysics.

Thanks for the suggestion of structural experiment, single channel experiment, and subunit assessment. However, structural analysis can not reveal trapped ketamine in the channel in the *in vivo* condition since it requires using proteins purified from *in vitro* cultured cell lines. Single-channel recording or subunit assessment can reveal microscopic properties of individual channel or channel composition, but they do not provide more information in terms of drug trapping. Below we provide the control experiments requested, and address your comments in point-by-point responses.

Major points

1. The authors employ a model of depression, the CRS, which they have used throughout their previous papers (Yang et al., 2018). The depressive-like phenotype varies across models (stronger anhedonia or behavioural despair depending on the model). Indeed, the authors observe quite of a large variance in the antidepressant efficacy at both 1h and 24h in both FST and SPT. This leads to rather weak statistical effects. Do the authors have any relationship appearing with decay of Ketamine and behaviour? Such a bulk analysis not taking into account individual variability remains outdated compared to the behavioural assertions available with new neurotechnologies (Cerniauskas et al., 2018). The authors should provide evidence for similar timelines of action in another model of depression, which if possible can have better face validity with the human condition.

The reviewer has a good point about tracking the relationship of decay time of ketamine and the behavioral performance at the individual level. However, such experiment would require measuring behavior of the same animal at different time points. Such practice is not recommended for behavioral measure since the tests at the earlier time point will affect the performance at the later time points. That is the exact reason why we chose to use different groups of animals for each time point. The statistics we presented in the paper represents data from several batches

of animals and indeed shows significance. Regarding the reviewer's request on a second model of depression, we have now used the WKY rat, which has been demonstrated to be a genetic model of depression^{1,2}. As shown in Figure R1 [REDACTED], in the FST assays, the antidepressant effects of ketamine on WKY rats are significant at both the 1 h and 24 h time points. This set of data further confirms ketamine's sustained effects.

[REDACTED]

Minor point. The authors miss some valuable controls for their behavioural assessment.

What

happens with treatment non-stress animals?

Thanks for the suggestion. We have now tested the effects of ketamine on naïve mice. We found that ketamine (i.p. 10 mg kg⁻¹) decreased immobile time in the FST at 1 h but not 24 h after injection (Figure R2a-c). This can be explained by our use-dependent trapping model of ketamine: the acute swim stress transiently induces Lhb burst firing and opens a portion of NMDARs which are blocked by ketamine. However, the number of opened NMDARs by the acute stress is much less than those in the depressive-like state induced by chronic stress. Therefore the effect is no longer significant at 24 h in the naïve mice. In the SPT test on naïve mice, there is no acute stress therefore ketamine has no effect even at 1 h (Figure R2d).

Figure R2. Antidepressant effects of ketamine in naïve mice.

a, Schematic of experimental paradigm for behavioral tests after i.p. injection of 10 mg kg⁻¹ ketamine in naïve mice.

b,c, The forced swim test (FST) at 1 h (**b**) and 24 h (**c**) after a single i.p. injection of ketamine on naïve mice ($P = 0.024$, $n=6, 7$ at 1 h; $P = 0.83$, $n = 8, 8$ at 24 h).

d, The sucrose preference test (SPT) at 1h after ketamine on naïve mice ($P = 0.76$, $n = 11, 11$) Sal: saline. Ket: ketamine. Unpaired t test.

* $P < 0.05$; NS, not significant. Error bars indicate SEMs.

2. The analysis of the burst is interesting however only expands previous findings. In structures including thalamus or VTA the burst activity relies on the RMP of neurons. The example shown indicate a 10mV difference in RMP. Is it the case that systematically cells bursting are more hyperpolarized? A better biophysical report of parameters should be provided.

The reviewer is totally correct that bursting cells in the LHB are more hyperpolarized than non-bursting neurons. This and other detailed analysis of bursting-related biophysical parameters were reported in Fig. 2 and Extended Fig. 6 of our 2018 paper³ (see figures below). In the current manuscript, we focus on analyzing the LHB burst at different time points, which stands as an important foundation for the mechanistic understanding of ketamine's sustained effects.

Figure R3. Bursting-related biophysical parameters (from Fig. 2d, 2e and Extended Fig. 6a-d in Yang et al.³)

When reporting the signatures of burst firing the authors miss to provide the control group. Is the CRS leading to increase bursting. This remains unknown and all data seem to relate only to the ketamine effect on the saline in CRS. Such a control is also missing from the *in vivo* data.

The increase of LHB burst both *in vitro* and *in vivo* in the CRS group has

been reported in our 2018 paper³ in Extended Data Fig. 3g-j and Extended Data Fig. 4e-g (see figures below). Consistent with these previous results, in the current manuscript, we also reported that bursting of the CRS group (Fig. 1f) is significantly higher than that of the naïve control as shown in Extended Data Fig. 2a ($P < 0.001$, Chi-Square test). These results were described as “The proportion of bursting neurons was significantly higher in CRS mice compared with naïve mice” (Page 5, Line 111-112 in previous version and Page 5, Line 112-113 in revision).

Figure R4. Chronic restraint stress increases burst firing in LHb (from Extended Fig. 3g-j and Extended Fig. 4e-g in Yang et al.³)

Minor point. The burst signatures provided in the methods differ from previous work published by the authors, thus this makes difficult to compare across literature. Is there a reason to define it in a different manner?

We thank the reviewer for the thoughtful comment. In the current manuscript, the onset of a burst remains as two spikes with a maximal inter-spike interval of 20 ms. For the ending of each burst, we changed the cutoff threshold of maximal inter-spike interval from 100 ms in the previous paper to the current 50 ms. Previous papers studying burst firing have used either 100 ms⁴⁻⁶ or 50 ms⁷⁻⁹ as the cutoff threshold for burst ending. Reason behind our change is that as we accumulate more *in vivo* recording data, if we continued to use 100 ms as burst ending threshold, occasionally we got extreme numbers in the “spike number per burst” parameter (>100 spikes per burst, see red arrow in Figure R5j). Therefore we adjusted the criterion to 50 ms. Below we compared major parameters before and after this change (Figure R5). The major conclusions, that

ketamine decreased bursting in the LHb *in vivo* 24 h after injection, is unchanged.

Figure R5. Comparison of burst results using old and new criteria.
a-d and f-i, Bar graphs illustrating the bursting spike frequency (a, f), bursts per minute (b, g), burst spike percentage (c, h) and spike number per burst (d, i) in CRS mice 24 h after saline or ketamine i.p. injection using current criteria (a-d) and previous criteria (f-i). n = 72, 86 units in 7 mice in saline group and 89, 92 units in 10 mice in ketamine group.
e and j, Histogram distribution of spike number per burst of all units in (d) and (i). Pie graph showing percentage of neurons with spike number per burst larger than 30.
 ** $P < 0.01$; *** $P < 0.001$; NS, not significant. Error bars indicate SEMs.

3. The authors have previously shown that burst are regulated by a complex machine that integrates NMDA receptors, calcium channels and even astrocytes. The authors should show specificity for the effect of Ketamine on NMDA and understand whether the other components are similarly affected.

Ketamine's direct molecular target(s) have been previously heavily studied¹⁰. Among a large number of molecules examined, NMDAR is the one with the highest binding affinity to ketamine¹⁰. There is no evidence that ketamine can bind the T type calcium channels and astrocytic K channel. That is why we focused our study on NMDAR. As to the specificity issue, as shown in the Figure R6 below, ketamine does not affect the AMPAR or GABA_AR components.

The measurement of NMDA and AMPA receptors currents and their analysis of I/O curves of absolute value is likely not appropriate. In many instances the currents seem also in the nA range which is at the mercy of space clamp issues. There is no doubt that ketamine can reduce NMDA currents as shown in a variety of studies throughout the brain, however the authors

should consolidate these findings in order to eliminate the potential issues of stimulus intensity and location which is difficult to reproduce across recordings. This is particularly the case in a structure that has not an organized anatomical architecture. More experiments, in better controlled conditions should be provided.

We totally agree with the reviewer that the absolute value of NMDAR or AMPAR currents alone may be affected by the stimulus intensity and location of the stimulating electrode. That is why we also additionally measured the ratio of NMDA/AMPA (N/A ratio) in Extended Figure 4 (now moved to Fig. 2d, 2j). This ratio normalizes the variability of stimulus intensity and location, and is standardly used in the field to quantify NMDAR or AMPAR responses¹¹⁻¹⁵. Importantly, our N/A ratio result also reveals consistent change of NMDAR response by ketamine at the 24 h point. We have readjusted the panels of Fig 2 and Extended Fig 4. The text is also adjusted accordingly (Page 7, Line156-174).

Regarding the reviewer's concern on space clamp issue, please note that all our recordings of NMDAR-EPSCs are in the pA range, not the nA range.

Majorly the NMDA component is assessed in this work. Yet, the authors have shown that hypothalamic stimulation strongly affects the AMPA component. Furthermore, the group of R. Malinow has shown the GABA component being affected in a model of depression. The authors need to provide solid evidence for the specificity of their effects.

Since ketamine is a well known inhibitor of NMDAR, not of AMPAR or GABAR¹⁰, we focused our study on NMDA component. Per the reviewer's request, we now also tested ketamine's effects on AMPAR or GABA_AR in LHb neurons and confirmed that these two components were not affected (Figure R6).

Figure R6. Ketamine does not alter the LHb AMPAR-eEPSCs and GABA_A-eIPSCs.
a, AMPAR-eEPSCs (normalized by baseline) during incubation and wash-out of ketamine (10 μM). AMPAR-eEPSCs are isolated by application of PTX and AP5 in ACSF under voltage clamp at -70 mV. n = 5 cells.
b, GABA_A-eIPSCs (normalized by baseline) during incubation and wash-out of ketamine (10 μM). GABA_A-eIPSCs are isolated by application of NBQX and AP5 in ACSF under voltage clamp at 0 mV. n = 5 cells.
 Error bars indicate SEMs.

4. The trapping hypothesis proposed by the authors is very intriguing. However, the data fall short to causally demonstrate that ketamine is really trapped within the channel. Furthermore, the trapping idea, as also indicated in their text was previously proposed in other work. This questions the degree of novelty of the study.

We certainly agree that the trapping property of ketamine as a NMDAR blocker has been well documented¹⁶⁻¹⁸ and have cited the relevant literatures. However, to our knowledge, nobody has ever proposed that such trapping property can be an important player in ketamine's long-term effect. Thus through this work, we reveal a mechanism hidden in plain sight where a simple molecular biophysical property explains an important therapeutic function of ketamine. Furthermore, building on this, we came up with strategies to either shorten or extend ketamine's sustained antidepressant effects *in vivo*. As pointed out by Reviewer 2, such study should be of broad interest to medicine, neuroscience, and biophysics.

What is the rationale in these experiments to use the 100uM dose in Figure 3. The authors nicely show in the initial figures that the real detection of Ketamine was in the order of 16uM.

This should be the concentration employed for these experiments as also done in Figure 4. Control experiments monitoring AMPA and GABA_A receptors currents are appropriate in this case. An appropriate control showing that Dyngo4a is an efficient blocker of endocytosis of NMDA should be shown as the author cite work using Dynosore rather and in a recombinant system.

We thank the reviewer for this comment. Since peak concentration of memantine can reach close to 100 μ M *in vivo*¹⁹, when comparing the effects of these two drugs in the *in vitro* experiment in Fig. 3a, we decided to first use the same 100 μ M dosage for the two drugs. This result demonstrates that at the same concentration, the two drugs behave differently. Next, exactly to address the reviewer's concern, and to use a dosage close to *in vivo* ketamine concentration, we lowered the ketamine concentration to 10 μ M in Fig. 4, and confirmed that at this *in-vivo*-relevant concentration, ketamine is still trapped for the recorded period.

As demonstrated in Figure R6 above, we now monitored the AMPAR and GABA_AR currents during the same recording period, and found them unaltered by ketamine.

To confirm that Dyngo-4a is an efficient blocker of endocytosis of NMDAR, we now tested the effect of Dyngo-4a in a protocol that induces endocytosis of NMDARs – the low-frequency stimulation (LFS)-induced long-term depression (LTD) of NMDAR on brain slices²⁰ (Figure R7, new Extended Fig. 6). Without Dyngo-4a, LFS successfully induced LTD of NMDAR-eEPSCs (Figure R7a, black dots). In presence of Dyngo-4a, however NMDAR-LTD was strongly suppressed (Figure R7a, red dots), suggesting that Dyngo-4a efficiently blocks endocytosis of NMDAR. We further showed that the NMDAR-eEPSCs can not recover after either 100 μ M or 10 μ M ketamine washout in the presence of the same concentration of Dyngo-4a (Figure R7b-e). Together these results suggest that the persistent suppression of NMDAR-eEPSCs after ketamine washout is not due to endocytosis. These results have been added to the revised manuscript as Extended Data Fig. 6. The text is also adjusted accordingly (Page 10, Line 215-222).

Figure R7. Recovery of Lhb NMDAR-eEPSCs after washout of ketamine in presence of endocytosis blocker Dyngo-4a (30 μM).

a, Dyngo-4a successfully blocks LFS (1Hz, 15min)-induced, endocytosis-based long-term depression of NMDAR-eEPSCs in hippocampal CA1 neurons. NMDAR-eEPSCs are isolated by application of PTX and NBQX in Mg^{2+} free ACSF under voltage clamp at -70 mV. In the Dyngo-4a group, 30 μM Dyngo-4a is additionally added in ACSF throughout the recording. $n = 7$ cells for vehicle group $n = 9$ cells for Dynago-4a group.

b, d, In presence of Dyngo-4a, Lhb NMDAR-eEPSCs (normalized by baseline) still do not show recovery after wash-out of 100 μM ketamine (**b**) or 10 μM ketamine (**d**) in Lhb neurons.

c, e, Bar graphs showing NMDAR-eEPSCs at the end of the 10 min perfusion period (left) and at 50-60 min (right). 100 μM : $n = 6$ cells for vehicle groups and $n = 5$ cells for Dynago-4a groups. 10 μM : $n = 6$ cells for vehicle groups and $n = 7$ cells for Dynago-4a groups. Ket: ketamine. At 10 min: vehicle vs Dyngo-4a, $P = 0.70$ in (**c**) and $P = 0.09$ in (**e**). At 50-60 min: vehicle vs Dyngo-4a, $P = 0.55$ in (**c**) and $P = 0.51$ in (**e**). The data of vehicle group in (**b**) was from the data of ket- group in Figure 3a and the date of vehicle group in (**d**) was from the data of “no stim” group in Figure 4a. Unpaired t test. NS, not significant. Error bars indicate SEMs.

The authors claim in Figure 4: These results provide a plausible mechanism for the prolonged blockade of NMDARs after ketamine clearance in plasma. Here, the authors fail to explain this statement. The demonstration that endocytosis is not part of the mechanism is poor (controls missing), there is no alternative strategy tested and the trapping hypothesis remains simply a preference. The authors cite a paper that does not support the trapping hypothesis but rather

promote the idea that lateral trafficking contribute to the maintenance of NMDA receptors pool. Altogether this renders this work quite preliminary, and descriptive likely appropriate to more specialized audience.

We thank the reviewer for the comments. Firstly, we have now provided the requested control experiment in the above reply and Figure R7 to rule out the endocytosis mechanism. Secondly, regarding the lateral trafficking, the reviewer is right that the cited paper from the Westbrook group suggests that lateral trafficking could contribute to the recovery of NMDAR-eEPSCs after ketamine inhibition in the hippocampal CA1 neurons. However, in the LHb, there is no such recovery. We have performed a series of experiments comparing this property of LHb and CA1 neurons in a separate manuscript, and now provide the results below.

[REDACTED]

[REDACTED]

5. The kick off protocol is intriguing. Also in this case a major control is missing. If the slice is not bathed in ketamine but in saline, and the kick off protocol is provided, is the NMDA component affected. There are instances of plasticity of NMDA in the hippocampus, the VTA and the habenula. This is a major drawback for the claims made by the authors. Is it possible that only postsynaptic activity represents a kick off state? If so, one would think that the conditions to kick off ketamine are larger in a depression state as the activity is higher. This should be tested experimentally.

We have now performed the important control experiment that the reviewer suggested, which is to test the kick off protocol on LHb neurons bathed in ACSF without ketamine treatment. We found no change in the NMDAR-eEPSCs (Figure R9, new Extended Fig. 8), therefore excluding contribution of plasticity mechanism. The text is adjusted accordingly (Page 12, Line 258-260).

Figure R9. “Kick off” protocol does not induce NMDAR-LTP. NMDAR-eEPSCs (normalized by baseline) after kick off protocol. NMDAR-eEPSCs are isolated by application of PTX and NBQX in Mg^{2+} free ACSF under voltage clamp at -70 mV. $n = 7$ cells. Error bars indicate SEMs.

Postsynaptic activity alone is not sufficient for kick-off. Presynaptic glutamate release is also necessary for receptor opening and drug release. Regarding the kick off under the depression state, since the activity is blocked after ketamine treatment, the kick off level would not necessarily be higher than naïve state. On the other hand, in an experiment we prepare for another manuscript, we found that the level of LHb-NMDAR blockade by ketamine is higher in a depressive-like state than in the naïve state (Figure R10).

[REDACTED]

*Why moving to a complete different protocol using higher stimulation in vivo?
These
presynaptic manipulations can engage plasticity mechanisms at many sites, and this was
not
controlled for. What the effect of this stimulation with Chrimson at synapses remains
completely unknown.*

As explained in our manuscript (Page12, Line 259-261 in previous version and Page 12 Line 266-268 in revision), to open NMDARs through voltage-clamp and zero Mg²⁺ condition is only plausible *in vitro*. Therefore we had to switch to a plausible method that can activate LHb neurons and open their NMDARs *in vivo*. Activation of a major presynaptic input into the LHb is such a method. For the concern of potential contribution from a plasticity mechanism, our saline-treated control groups did not show difference in either the NMDAR-eEPSCs (Fig. 5g, 5k) or AMPAR-eEPSCs (Figure R11) between the mCherry- and ChrimsonR- expressing groups.

This suggests that a photostimulation-triggered plasticity mechanism is unlikely to contribute to the altered NMDAR-eEPSCs in the ketamine-treated groups.

Figure R11. AMPAR currents of Lhb neuron are not altered after photostimulation.

a, AMPAR-eEPSCs of Lhb neurons recorded at 24 h after ketamine or saline injection from experiment in Figure. 5f. ($P = 0.06$, CRS-sal-mCherry vs CRS-sal-ChrimsonR, Mann-Whitney test). $n = 36$ cells in 3 mice in CRS-sal-mCherry group, $n = 40$ cells in 4 mice in CRS-ket-mCherry group, $n = 34$ cells in 3 mice in CRS-sal-ChrimsonR group, $n = 38$ cells in 4 mice in CRS-ket-ChrimsonR group.

b, AMPAR-eEPSCs of Lhb neurons recorded at 24 h after ketamine or saline injection from experiment in Figure. 5j ($P = 0.41$, CRS-sal-mCherry vs CRS-sal-ChrimsonR, Mann-Whitney test). $n = 34$ cells in 4 mice in CRS-sal-mCherry group, $n = 35$ cells in 3 mice in CRS-ket-mCherry group, $n = 27$ cells in 3 mice in CRS-sal-ChrimsonR group, $n = 29$ cells in 3 mice in CRS-ket-ChrimsonR group.

NS, not significant. Error bars indicate SEMs.

6. The choice and rationale for choosing the hypothalamic input is not clear. The authors published recently the importance of this pathway in stress driven depression. This once again reduces the novelty of the presented work. It would have been instead useful to provide information on the efficiency of multiple inputs onto habenula. The LH is only one among many releasing glutamate. Is the hypothesis valid for every afferent that is excitatory?

We respectfully disagree that using a published major Lhb input as a method to achieve Lhb stimulation would reduce the novelty of the current manuscript. The goal here is to have a strong input and activate as many as Lhb neurons as possible. As demonstrated in our previous paper, LH is the strongest one among eight Lhb input pathways tested. When the LH input was optogenetically stimulated, 100% Lhb neurons were activated²¹, making LH an ideal pathway for our manipulation experiment. We would like to emphasize that the experiment here is not to claim the LH as the only endogenous pathway capable of kicking off ketamine, but to use its activity as a stimulation tool to test our hypothesis on ketamine trapping. Therefore we think that testing every afferent pathway is not only beyond the scope of the current manuscript, but would likely defer the purpose by introducing less effective input(s).

Minor point. The optogenetic Chrimson-based experiment lacks some controls. What happens if the authors swap the concentrations in vivo, and run the same timed stimulation protocols?

It remains unclear why the model works not knowing the stoichiometry of NMDA receptors at every different stage of ketamine presence. Is the Kon-Koff model proposed by the authors better to be tested in heterologous systems enabling the choice of specific subunits. This would provide better information as this basic knowledge is missing in this neuronal circuit.

We thank the reviewer for the suggestions. For the experiments in Fig. 5f-h, we hoped to test whether ketamine's sustained effects would be shortened by activity kick-off. If we used a lower dosage (5 mg kg⁻¹) here, there would not have been a sustained effect at 24 h to start with. Likewise, for the experiments in Fig 5j-l, we hoped to test whether more trapping could extend ketamine's effects. If we had used a high dosage (10 mg kg⁻¹), it would already have caused sustained effect at 24 h. Therefore, we could not swap the concentrations for these two experiments.

The Mameli group has made nice attempts to determine the stoichiometry of NMDA receptors in the LHb by using NR2A and NR2B blocker²². There was some complication in the result since the blocked NMDAR currents by both blockers was even smaller than those blocked by one blocker alone. Therefore it was difficult to derive the stoichiometry from the subunit blocker experiment. In addition, we have measured the total NMDAR response at different stage of ketamine presence. This is a more direct and relevant evidence than the stoichiometry information, since our model is about NMDAR blockade.

As discussed in the manuscript, the K_{on}-K_{off} dynamics can be very different in the *in vitro* heterologous system, where the agonist is applied for a much longer duration (second range) than the real presence of glutamate in the *in vivo* situation (millisecond range).

Minor point

In the introduction the authors state experimental demonstration for the repercussions of habenular bursting on downstream dopamine and serotonin systems. This should be toned down as for the moment it remains a non-tested hypothesis.

We thank the reviewer for the suggestion and have now added the word “potentially” in this statement (Page 3, Line 56).

The discussion would need likely some revision. The authors explain their view on how to leverage the trapping mechanisms in the context of therapeutics. This remains far stretched as it is not clear what is the vision of providing negative stimuli to patients. The authors previously indicate the importance of plasticity in the context of stress – now it seems that stress can be leveraged without consequences. This may lead to confusion rather than clarity in the field.

Stress is not always bad, and has been heavily documented to have a U-shape effect on cognition and psychological health²³. The approach we propose in the discussion utilizes controlled, mild stress within a specific time window. This is different from the chronic, uncontrollable stress that induces depression.

References

- 1 Will, C. C., Aird, F. & Redei, E. E. Selectively bred Wistar-Kyoto rats: an animal model of depression and hyper-responsiveness to antidepressants. *Mol Psychiatry* **8**, 925-932, doi:10.1038/sj.mp.4001345 (2003).
- 2 Willner, P. *et al.* Validation of chronic mild stress in the Wistar-Kyoto rat as an animal model of treatment-resistant depression. *Behavioural Pharmacology* **30**, 239-250, doi:10.1097/Fbp.0000000000000431 (2019).
- 3 Yang, Y. *et al.* Ketamine blocks bursting in the lateral habenula to rapidly relieve depression. *Nature* **554**, 317-322, doi:10.1038/nature25509 (2018).
- 4 Yuan, Y. *et al.* Reward Inhibits Paraventricular CRH Neurons to Relieve Stress. *Curr Biol* **29**, 1243-1251 e1244, doi:10.1016/j.cub.2019.02.048 (2019).
- 5 Li, H., Pullmann, D. & Zhou, T. C. Valence-encoding in the lateral habenula arises from the entopeduncular region. *Elife* **8**, doi:10.7554/eLife.41223 (2019).
- 6 Baek, J. *et al.* Neural circuits underlying a psychotherapeutic regimen for fear disorders. *Nature* **566**, 339-343, doi:10.1038/s41586-019-0931-y (2019).
- 7 Kim, M. J., Mizumori, S. J. & Bernstein, I. L. Neuronal representation of conditioned taste in the basolateral amygdala of rats. *Neurobiol Learn Mem* **93**, 406-414, doi:10.1016/j.nlm.2009.12.007 (2010).
- 8 Burgos-Robles, A., Vidal-Gonzalez, I., Santini, E. & Quirk, G. J. Consolidation of fear extinction requires NMDA receptor-dependent bursting in the ventromedial prefrontal cortex. *Neuron* **53**, 871-880, doi:10.1016/j.neuron.2007.02.021 (2007).
- 9 Weir, K., Blanquie, O., Kilb, W., Luhmann, H. J. & Sinning, A. Comparison of spike parameters from optically identified GABAergic and glutamatergic neurons in sparse cortical cultures. *Front Cell Neurosci* **8**, 460, doi:10.3389/fncel.2014.00460 (2014).
- 10 Zanos, P. *et al.* Ketamine and Ketamine Metabolite Pharmacology: Insights into Therapeutic Mechanisms. *Pharmacol Rev* **70**, 621-660, doi:10.1124/pr.117.015198 (2018).
- 11 Gao, Y. *et al.* beta2-microglobulin functions as an endogenous NMDAR antagonist to impair synaptic function. *Cell* **186**, 1026-1038 e1020, doi:10.1016/j.cell.2023.01.021 (2023).
- 12 Etherton, M. R., Tabuchi, K., Sharma, M., Ko, J. & Sudhof, T. C. An autism-associated point mutation in the neuroligin cytoplasmic tail selectively impairs AMPA receptor-mediated synaptic transmission in hippocampus. *EMBO J* **30**, 2908-2919, doi:10.1038/emboj.2011.182 (2011).
- 13 McCauley, J. P. *et al.* Circadian Modulation of Neurons and Astrocytes Controls Synaptic Plasticity in Hippocampal Area CA1. *Cell Rep* **33**, 108255, doi:10.1016/j.celrep.2020.108255 (2020).

- 14 Qiao, H., Foote, M., Graham, K., Wu, Y. & Zhou, Y. 14-3-3 proteins are required for hippocampal long-term potentiation and associative learning and memory. *J Neurosci* **34**, 4801-4808, doi:10.1523/JNEUROSCI.4393-13.2014 (2014).
- 15 Chang, F. Y., Lee, C. C., Huang, C. C. & Hsu, K. S. Unconjugated bilirubin exposure impairs hippocampal long-term synaptic plasticity. *PLoS One* **4**, e5876, doi:10.1371/journal.pone.0005876 (2009).
- 16 MacDonald, J. F., Miljkovic, Z. & Pennefather, P. Use-dependent block of excitatory amino acid currents in cultured neurons by ketamine. *J Neurophysiol* **58**, 251-266, doi:10.1152/jn.1987.58.2.251 (1987).
- 17 MacDonald, J. F. *et al.* Actions of ketamine, phencyclidine and MK-801 on NMDA receptor currents in cultured mouse hippocampal neurones. *J Physiol* **432**, 483-508, doi:10.1113/jphysiol.1991.sp018396 (1991).
- 18 Mealing, G. A., Lanthorn, T. H., Murray, C. L., Small, D. L. & Morley, P. Differences in degree of trapping of low-affinity uncompetitive N-methyl-D-aspartic acid receptor antagonists with similar kinetics of block. *J Pharmacol Exp Ther* **288**, 204-210 (1999).
- 19 Beconi MG, H. D., Park L, Lyons K, Giuliano J, Dominguez C, Munoz-Sanjuan I, Pacifici R. Pharmacokinetics of memantine in rats and mice. *PLOS Currents Huntington Disease* (2012).
- 20 Montgomery, J. M., Selcher, J. C., Hanson, J. E. & Madison, D. V. Dynamin-dependent NMDAR endocytosis during LTD and its dependence on synaptic state. *BMC Neurosci* **6**, 48, doi:10.1186/1471-2202-6-48 (2005).
- 21 Zheng, Z. *et al.* Hypothalamus-habenula potentiation encodes chronic stress experience and drives depression onset. *Neuron*, doi:10.1016/j.neuron.2022.01.011 (2022).
- 22 Nuno-Perez, A., Mondoloni, S., Tchenio, A., Lecca, S. & Mameli, M. Biophysical and synaptic properties of NMDA receptors in the lateral habenula. *Neuropharmacology* **196**, doi:ARTN 108718 10.1016/j.neuropharm.2021.108718 (2021).
- 23 Yuen, E. Y. *et al.* Repeated stress causes cognitive impairment by suppressing glutamate receptor expression and function in prefrontal cortex. *Neuron* **73**, 962-977, doi:10.1016/j.neuron.2011.12.033 (2012).

Referee #2:

Ma et al. A. Summary of key results: Ketamine is highly effective for treatment of depression, yet its mechanism of action is not understood: especially the paradox that the behavioral effects of ketamine last about 100 times longer than its half-life for elimination. This study presents the extraordinary finding that the prolonged action of ketamine beyond its tissue concentration lifetime is because of prolonged channel block of the NMDAR receptor by ketamine.

B. Originality and significance: To my knowledge the findings are original and highly significant. Now that the the mechanism of ketamine is known, as shown in this study to be prolonged channel block, then it should be possible to devise even more effective prolonged blocking agents. The biophysical observations that support the proposed mechanism of channel block

are backed up with appropriate behavioral studies to support the conclusion of prolonged block. This paper should be of wide interest in medicine, neuroscience, and biophysics.

C. Data and Methodology: The validity of the approach and the quality of the data support the conclusions drawn.

D. Statistics: Statistics are appropriate.

E. Conclusions: The conclusions are well supported by the presented data.

F. Improvements: I have no suggested improvements or experiments.

G. References: References have been made to previous work, but because I do not work directly in the fields of depression and ketamine, I would not know if all appropriate references have been made.

H. The paper is very well written and the title, abstract, introduction, results, and discussion are clear and appropriate for the conclusions.

We are very grateful to the reviewer for his/her strong enthusiasm and support! We now address the comments below, and modify the figures and text correspondingly.

No major suggestions.

Minor suggestions are below where the numbers indicate line numbers.

1. Title: If there is room, spell out LHb

Thank you for the suggestion. We now change the title to “Sustained Antidepressant Effect of Ketamine Through NMDAR Trapping in Lateral Habenula”.

68. or the ketamine metabolite

Corrected.

99. because the apparent off rate of ketamine is so slow, it seems that 0.23 uM might still contribute to binding. Comment.

Yes. This is indeed the point we would like to make. Because we discovered in this manuscript that the apparent off rate of ketamine is very slow, it can still inhibit the channel at a concentration much lower than its IC₅₀. 0.23 μM is the brain concentration at 1 h after injection. At 24 h after injection, the concentration has become undetectable.

107. *There is no e and f in Fig. 1*

149. *fig. 4 = Fig. 4*

160. *fig. = Fig*

We have corrected these accordingly.

Extended Fig. 8. Rather than repeat the text after the title, perhaps the title should be a conclusion. i.e. Ketamine concentration in the brain falls exponentially with a half time of 12?? minutes or what ever it is.

Thank you for the suggestion. We have combined the title and following text into one sentence “Brain concentration of ketamine after a single i.p injection of 5 mg kg⁻¹ ketamine in CRS mice, as measured by LC-MS/MS.”

280. *folds should be fold*

Corrected.

288 An brief explanation of mCherry and ChrimsonR expressing mice would be useful here to interpret the findings.

We have added a brief explanation as below: In control mCherry-expressing mice, which do not have light-induced activity in the LHb,.... In ChrimsonR-expressing mice, which have light-induced activity in the LHb.... (Page 13, Line 295 and Line 297-298).

331 suggest: and illustrate a case where a biophysical channel block mechanism extended to extraordinarily long times by the in vivo physiological parameters explains an import therapeutic function. (I suspect you can write a better version.)

Thank you for the great suggestion! We have changed the sentence to your suggested version (Page15, Line 338-340).

370 *fig. = Fig*

Corrected.

924. Laser stimulation (put some brief details here of what laser stimulation was).

We have added the detailed information as: 635 nm, 40 Hz, 2 ms pulse, 250 μW (Page 36, Line 980-981).

Reviewer Reports on the First Revision:

Referees' comments:

Referee #1 (Remarks to the Author):

In this revised manuscript the authors addressed all the concerns and points raised during the reviewing process. This piece of work adds important cellular information on the effects produced by ketamine and I thank the authors for addressing the points I mentioned both with constructive discussions and new experiments.

Referee #2 (Remarks to the Author):

A-H are unchanged from my previous review.

The authors have suitably addressed my previous review comments.

Additional minor review comments.

Mention "mice" somewhere at the beginning of the Discussion. Perhaps ". . . for up to 24 h in mice (Fig. 1, 2).

The data in Figure 1 supports the title in its legend. However Extended Data Figure 1 shows controls, so the data in Extended Data Figure 1 does not support the legend title, which is confusing. Consequently, the title for Extended Data Figure 1 should be followed by a sentence indicating what is actually shown or the title should be changed.

The Extended Data Figure 10 legend may be missing some lines. It needs n values and SEM or SD as appropriate and also indicate the half life in minutes.

Referee #3 (Remarks to the Author):

In this manuscript, the authors aim to reveal the mechanism by which ketamine (but not memantine) exerts its sustained antidepressive effect in people suffering from treatment resistant depression. They use as a model system, mice subjected to 14 days of chronic restraint stress and perform behavioral and electrophysiological tests.

Previously, this group showed that antidepressant effect of KET in mice correlates with immediate inhibition of bursting behavior in neurons of the lateral habenula (LHb) neurons (Yang 2018). Here, they investigate the effect of KET at later time points after administration. In Fig 1 they measure the time course for installation and maintenance of antidepressive effects of one KET injection (10 mgs/kg) and they confirm that, as for human patients, depressive-like behaviors (in this case FST,

and SPT) in mice decline rapidly (within one hour) and remain suppressed (sustained effect) for at least 24 hours. While this time course mirrors what has been reported for human patients the concentrations used are of concern. Specifically, the half-dose for antidepressant effect observed here is 8 μM (Fig 1a), which corresponds to full anesthesia in humans (Little et al., 1972; Idvall et al., 1979; Grant et al., 1983; with steady state unconsciousness for levels of 9 μM as reviewed in Zanos 2018).

Nevertheless, the authors observe that, as for human patients, although the KET concentration in brain declines rapidly, the antidepressive effect persists at the 24 hr point. This begs the question: if KET was eliminated from plasma within minutes, how can it still exert antidepressive action at 24 hours. This is a long-standing and vexing question in the field and in this manuscript the authors do a good job pointing to the urgency of answering it. They also enumerate the multiple answers that other groups have put forth over the past decade, which NMDA receptor mediated plasticity, and ketamine metabolites (HNK) . Here, the authors concentrate on the firing activity in lateral habenula, a brain area where they have long-standing expertise. Specifically, they test their previous hypothesis that antidepressant effect of KET is mediated by decreased firing of LHB neurons, which is mediated by NMDA receptor inhibition.

The authors reported previously that increased bursting activity in this brain region correlates with behavioral depression and the bursting activity can be reduced by NMDA receptor antagonists (ketamine or AP5) but not AMPA receptor antagonists or fluoxetine, a SSRI antidepressant (Yang 2018). In a companion paper (Cui 2018), this group reported that bursting of LHB neurons depends on inputs from (at least) astroglia. Therefore, the activity of the habenula is regulated by extrinsic factors.

However, these previous studies did not address a critical point: why only ketamine but not other NMDA receptor inhibitors (such as AP5), which also reduce bursting of the LHB neurons, do not relieve depression? In the present manuscript they compare ketamine and memantine and found that only ketamine produces long-lasting (1 hour) inhibition of NMDA receptors, whereas memantine washes off within 10 minutes. This is an interesting result. Unfortunately, these experiments are done with 100 μM of either KET or MEM, doses that are 100-fold higher than their IC_{50} for NMDA receptors, and 10-fold larger than the concentration of KET used in anesthesia (10 μM). Nevertheless, the result is interesting and the authors go on to show that following inhibition with 10 μM KET, NMDA receptor activity can be recovered by stimulating neural activity. Which prompts them to propose that the long-lasting effect of KET on inhibiting NMDA receptors is due to 'trapping block'. Here the paper veers away from the previous rigor and into unsubstantiated hypotheses.

So far, the existing evidence supports similar mechanisms of action for both memantine and ketamine: open-channel block (with similar IC_{50} ~1 μM) (mostly from Jon Johnson's work; Glasgow 2018); and this is supported by structural studies (Zhang 2021 - the Zhu group, Chou 2022 – the Furukawa group). The observation that the ketamine inhibition of NMDA receptor-mediated bursting in LHB can be relieved by stimulation is an observation consistent with persistent open-channel block, but does not come close to a mechanism, or to explaining why KET becomes trapped and MEM does not. One is left to wonder whether this is specific to NMDA receptors endogenous to LHB or to NMDA receptors on neurons that innervate the LHB, or some other mechanism. I must concur with Reviewer 1 that to claim trapping mechanism, one must show experiments with recombinant receptors of distinct composition, single channel recordings from LHB neurons, and some indication that KET remains bound to receptors for long durations (hours!!) is necessary.

In my view, the results presented here represent additional evidence for their previous hypothesis that inhibition of burst firing in LHb correlates with cessation of depressive behaviors.

Additional comments from Referee #2:

Reviewer 3's comment is that: "I must concur with Reviewer 1 that to claim trapping mechanism, one must show experiments with recombinant receptors of distinct composition, single channel recordings from LHb neurons, and some indication that KET remains bound to receptors for long durations (hours!!) is necessary.

Reviewer 3 requests evidence at the single channel level for channels of known composition to show long term trapping. The authors did not present such single channel data, but used evoked eEPSCs as an assay to determine the relative percentage of blocked and unblocked NMDARs and also behavioral studies to suggest long term effects after wash out. .

The authors less direct approach was consistent with long term trapping and was sufficient for me but not for Reviewer 1 initially and not for Reviewer 3. Note that Reviewer #1 changed their opinion after the authors revised the paper and presented additional data in their response for the revised paper. The question is not whether the authors did the types of experiments the reviewers might like in an isolated model system, but whether the experiments they did in brain and brain slices support their hypothesis of long term trapping, which, in my opinion, they do. Do they prove long term trapping? No. Are scientific questions ever proved? No. The data are either consistent or inconsistent with the hypothesis, and the authors findings support long term trapping in my opinion, but I am not an expert on NMDARs. I am a channel biophysics type.

Brief summary of some of the experiments the authors have presented to support prolonged trapping of ketamine (ket) in NMDAR channels.

Fig. 1 shows prolonged behavioral effects of ket far beyond the wash out times, assuming the statistics in Fig. 1 k and L are significant which needs to be checked (see statistics below). Prolonged channel block by ket would be consistent with this observation.

Fig. 2 shows prolonged effects of ket on reducing the NMDA (but not AMPA) currents in slices. Prolonged channel block of NMDAR by ket would be consistent with this observation.

Fig. 3 (100 uM ket or memantine (mem) show that after 10 minute exposure both drugs reduce the NMDAR eEPSCs but that there is recovery from the block over the next hour for mem but not recovery over the next hour for ket, just as would be expected if mem unblocked faster than ket based on previous studies. If the prolonged effects were due to ket and mem being slowly released from reservoirs in the tissue and then reblocking the NMDAR channels, then the results from ket and mem might have been expected to mimic each other, but they did not.

Fig. 4 shows that a protocol to open the NMDR channels that would presumably release ket from channel block reduces the inhibition of the NMDAR eEPSCs JUST AS WOULD BE EXPECTED IF KET WERE A VERY LONG LASTING CHANNEL BLOCKER.

Fig. 5 shows that an experiment that presumably would untrap ket from NMDA receptors at one hr results in no NMDAR eEPSC reduction at 24 hrs (Fig. 5 d-h), whereas an experiment that would presumably trap additional ket at 10 min leads to reduction in NMDAR eEPSCs at 24 hrs (Fig. 5 i-l), as would be expected if ket were a very long lasting channel blocker.

The authors have presented data to support their hypothesis of long term trapping in real brain and brain slices. Are other explanations possible? Other explanations are always possible in such complex systems, but if one wants to study native receptors in real cells in real brain or brain slices, then the experiments are by necessity more complex in design and interpretation.

Yes, experiments will have to be done in model systems down the road as suggested by Reviewer #3, but these are different types of experiments for a different study.

In light of the concerns of Reviewer #3 I would suggest that the authors be more tempered in their conclusions (and in the title as well) indicating that their findings are consistent with (support) an hypothesis of long term trapping and do not directly establish long term trapping.

Also, in the discussion The authors might also consider the concept of whether delayed diffusion [B. Katz et al. J Physiol. 1973 The binding of acetylcholine to receptors and its removal from the synaptic cleft] might or might not play a role. Basically, as KET is released from NMDARs and diffuses out of the synaptic clefts it could likely rebind to unbound NMDARs receptors multiple times greatly extending its effective duration of action. Hence, a binding time of a few hours might be extended to days of action, and observed single channel binding times could be effectively extended many fold in the synaptic cleft through delayed diffusion when not observed in single channel

Statistics: In re-reading the paper I found some potential problems with the statistics in a few parts of some figures which I list below that need to be checked by the authors. Of course these concerns could be errors in my measurements and calculations.

A rule of thumb for SEM error bars and unpaired t-tests is that if the SEM error bars overlap or are separated by short distances for the two data sets to be compared, then the differences in means are unlikely to be significant. This can be seen in Fig. 2 where there needs to be gaps between the error bars of the compared data sets to obtain significance, and large gaps to obtain higher significance.

In Fig. 1 K the right most panel for ket blue the undrawn downward error bar would overlap with the upward error bar. I calculate a $P \sim 0.25$ (not significant) for this while the authors calculate a P of: $*** < 0.001$, highly significant. (I used a two tailed t-test) This needs to be checked, and of course I may have mis-calculated.

In Fig. 1 K ket middle panel for comparison of baseline and 0-5 min I get a P of about 0.04, not < 0.0001 as the authors get. The other 0 to 1 hr significance levels are also suspect.

In Fig. 1 L middle and right there appear to be similar problems suggesting that the actual P values may be considerably greater than the indicated P values.

At the end of the Fig. 1 legend, the “n = 72, 86 units . . . and 89, 92 units” needs to be explained or corrected in terms of n and units. . Why are n and units values different?

Fig. 4b. For plot to the left of Fig. 4b, state directly if the error bars in this plot come from the 5 and 6 separate cells mentioned for n. State if there is only one 60 minute run for each cell. In Fig. 4b the observed plotted points do not appear to match the stated n's unless there is perfect overlap of some plotted points. If so move overlapping points laterally so they can be seen. Place the SEM bars in Fig. 4b as this is stated as an unpaired t-test. Check the P values in Fig. 4b.

ALL FIGURES NEED SEM BARS if t-tests are applied to the data and all P values need to be checked if the P values do not appear consistent with the SEM bars.

Extended data Figure 3 d. The Fig. legend mentions SEM error bars but there are no error bars in this figure. State the test for part d.

Author Rebuttals to First Revision:

Referee #2 (Remarks to the Author):

A-H are unchanged from my previous review.

The authors have suitably addressed my previous review comments.

We are immensely grateful to reviewer 2 for his/her continuous support.

Additional minor review comments.

Mention "mice" somewhere at the beginning of the Discussion. Perhaps ". . . for up to 24 h in mice (Fig. 1, 2).

Thank you. We now added "in mice" in this sentence.

The data in Figure 1 supports the title in its legend. However Extended Data Figure 1 shows controls, so the data in Extended Data Figure 1 does not support the legend title, which is confusing. Consequently, the title for Extended Data Figure 1 should be followed by a sentence indicating what is actually shown or the title should be changed.

Thank you for the suggestion. We now changed the legend title to "Single injection of ketamine no longer causes antidepressant effects on day 3 and day 7" (Page29, Line 784-785).

The Extended Data Figure 10 legend may be missing some lines. It needs n values and SEM or SD as appropriate and also indicate the half life in minutes.

We now added the SEM and indicated the half-life to be 13 min in the legend.

Referee #3 (Remarks to the Author):

In this manuscript, the authors aim to reveal the mechanism by which ketamine (but not memantine) exerts its sustained antidepressive effect in people suffering from treatment resistant depression. They use as a model system, mice subjected to 14 days of chronic restraint stress and perform behavioral and electrophysiological tests. Previously, this group showed that antidepressant effect of KET in mice correlates with immediate inhibition of bursting behavior in neurons of the lateral habenula (LHb) neurons (Yang 2018). Here, they investigate the effect of KET at later time points after administration. In Fig 1 they measure the time course for installation and maintenance of antidepressive effects of one KET injection (10 mgs/kg) and they confirm that, as for human patients, depressive-like behaviors (in this case FST, and

SPT) in mice decline rapidly (within one hour) and remain suppressed (sustained effect) for at least 24 hours. While this time course mirrors what has been reported for human patients the concentrations used are of concern. Specifically, the half-dose for antidepressant effect observed here is 8 μM (Fig 1a), which corresponds to full anesthesia in humans (Little et al., 1972; Idvall et al., 1979; Grant et al., 1983; with steady state unconsciousness for levels of 9 μM as reviewed in Zanos 2018).

We thank the reviewer for bringing up the issue of species differences in drug dosage. It is well known that due to different pharmacokinetics across species, drugs are most often applied at different concentrations in different species^{1,2}. For example, for its anesthesia function, ketamine is used at 1-2 mg/kg (corresponding to 60 μM peak concentration) in human³, and 100 mg/kg⁴ (corresponding to $\sim 63 \mu\text{M}$ peak concentration) in mice⁵. According to the guide for dose conversion between animals and human², using body surface area for dosage conversion, a 10 mg/kg dose in mice would be equivalent to a human dose of $\sim 0.8\text{mg/kg}$, which is close to ketamine's antidepressant dosage for human patients.

In addition, I would also like to point out that in Fig. 1a what we demonstrated is that ketamine's half-life is 13 min, when its concentration has dropped to 8 μM . The concentration only transiently reaches above 8 μM for the initial ~ 10 min and drops way below this concentration within 1 hr. Our slice experiment in Fig. 3 also tries to mimic this physiological range of drug dynamics by presenting 10 μM ketamine for only 10 min and then wash off the drug. In contrast, for human anesthesia, as shown in the following figure, ketamine concentration is maintained constantly at 10 μM for 2 hrs. Therefore, the antidepressant dose we use here, which is a commonly used antidepressant dose in mice⁶⁻⁹, is much lower than the anesthesia dose in human.

FIG. 2. Plasma concentrations of ketamine, nor-ketamine (metabolite I) and dehydro-nor-ketamine (metabolite II) during anaesthesia. Mean + SD. All anaesthetics ≥ 150 min.

From Idvall, J. et al.³

Nevertheless, the authors observe that, as for human patients, although the KET concentration in brain declines rapidly, the antidepressive effect persists at the 24 hr point. This begs the question: if KET was eliminated from plasma within minutes, how can it still exert antidepressive action at 24 hours. This is a long-standing and vexing question in the field and in this manuscript the authors do a good job pointing to the urgency of answering it. They also enumerate the multiple answers that other groups have put forth over the past decade, which NMDA receptor mediated plasticity, and ketamine metabolites (HNK). Here, the authors concentrate on the firing activity in lateral habenula, a brain area where they have long-standing expertise. Specifically, they test their previous hypothesis that antidepressant effect of KET is mediated by decreased firing of LHB neurons, which is mediated by NMDA receptor inhibition. The authors reported previously that increased bursting activity in this brain region correlates with behavioral depression and the bursting activity can be reduced by NMDA receptor antagonists (ketamine or AP5) but not AMPA receptor antagonists or fluoxetine, a SSRI antidepressant (Yang 2018). In a companion paper (Cui 2018), this group reported that bursting of LHB neurons depends on inputs from (at least) astroglia. Therefore, the activity of the habenula is regulated by extrinsic factors. However, these previous studies did not address a critical point: why only ketamine but not other NMDA receptor inhibitors (such as AP5), which also reduce bursting of the LHB neurons, do not relieve depression? In the present manuscript they compare ketamine and memantine and found that only ketamine produces long-lasting (1 hour) inhibition of NMDA receptors, whereas memantine washes off within 10 minutes. This is an interesting result. Unfortunately, these experiments are done with 100 μM of either KET or MEM, doses that are 100-fold higher than their IC_{50} for NMDA receptors, and 10-fold larger than the concentration of KET used in anesthesia (10 μM).

We thank the reviewer for this comment. As explained in our last round of rebuttal, we first chose 100 μM of KET or MEM for the wash-out experiment since with 10 mg kg^{-1} i.p. injection, peak concentration of memantine can reach close to 100 μM *in vivo*¹⁰. Therefore when comparing the effects of these two drugs in the *in vitro* experiment in Fig. 3a, we decided to first use the same 100 μM dosage for the two drugs. This result demonstrates that at the same concentration, the two drugs behave differently. Next, exactly to address the reviewer's concern, and to use a dosage close to *in vivo* ketamine concentration in mice, we lowered the ketamine concentration to 10 μM in Fig. 4, and confirmed that at this *in-vivo*-relevant concentration, ketamine is still trapped for the recorded period.

Nevertheless, the result is interesting and the authors go on to show that following inhibition with 10 μM KET, NMDA receptor activity can be recovered by stimulating neural activity. Which prompts them to propose that the long-lasting effect of KET on inhibiting NMDA receptors is due to 'trapping block'. Here the paper veers away from

the previous rigor and into unsubstantiated hypotheses. So far, the existing evidence supports similar mechanisms of action for both memantine and ketamine: open-channel block (with similar $IC_{50} \sim 1 \mu M$) (mostly from Jon Johnson's work; Glasgow 2018); and this is supported by structural studies (Zhang 2021 - the Zhu group, Chou 2022 – the Furukawa group). The observation that the ketamine inhibition of NMDA receptor-mediated bursting in LHB can be relieved by stimulation is an observation consistent with persistent open-channel block, but does not come close to a mechanism, or to explaining why KET becomes trapped and MEM does not. One is left to wonder whether this is specific to NMDA receptors endogenous to LHB or to NMDA receptors on neurons that innervate the LHB, or some other mechanism.

Our data suggests that both drugs are trapped, but ketamine has a much slower off rate. This is consistent with the previous report¹¹ that despite the same IC_{50} , ketamine has a slower off rate than memantine.

I must concur with Reviewer 1 that to claim trapping mechanism, one must show experiments with recombinant receptors of distinct composition, single channel recordings from LHB neurons, and some indication that KET remains bound to receptors for long durations (hours!!) is necessary.

In my view, the results presented here represent additional evidence for their previous hypothesis that inhibition of burst firing in LHB correlates with cessation of depressive behaviors.

Additional comments from Referee #2:

Reviewer 3's comment is that: "I must concur with Reviewer 1 that to claim trapping mechanism, one must show experiments with recombinant receptors of distinct composition, single channel recordings from LHB neurons, and some indication that KET remains bound to receptors for long durations (hours!!) is necessary.

Reviewer 3 requests evidence at the single channel level for channels of known composition to show long term trapping. The authors did not present such single channel data, but used evoked eEPSCs as an assay to determine the relative percentage of blocked and unblocked NMDARs and also behavioral studies to suggest long term effects after wash out. .

The authors less direct approach was consistent with long term trapping and was sufficient for me but not for Reviewer 1 initially and not for Reviewer 3. Note that Reviewer #1 changed their opinion after the authors revised the paper and presented additional data in their response for the revised paper. The question is not whether the authors did the types of experiments the reviewers might like in an isolated model system, but whether the experiments they did in brain and brain slices support their hypothesis of long term trapping, which, in my opinion, they do. Do they prove

long term trapping? No. Are scientific questions ever proved? No. The data are either consistent or inconsistent with the hypothesis, and the authors findings support long term trapping in my opinion, but I am not an expert on NMDARs. I am a channel biophysics type.

Brief summary of some of the experiments the authors have presented to support prolonged trapping of ketamine (ket) in NMDAR channels.

Fig. 1 shows prolonged behavioral effects of ket far beyond the wash out times, assuming the statistics in Fig. 1 k and L are significant which needs to be checked (see statistics below). Prolonged channel block by ket would be consistent with this observation.

Fig. 2 shows prolonged effects of ket on reducing the NMDA (but not AMPA) currents in slices. Prolonged channel block of NMDAR by ket would be consistent with this observation.

Fig. 3 (100 uM ket or memantine (mem) show that after 10 minute exposure both drugs reduce the NMDAR eEPSCs but that there is recovery from the block over the next hour for mem but not recovery over the next hour for ket, just as would be expected if mem unblocked faster than ket based on previous studies. If the prolonged effects were due to ket and mem being slowly released from reservoirs in the tissue and then reblocking the NMDAR channels, then the results from ket and mem might have been expected to mimic each other, but they did not.

Fig. 4 shows that a protocol to open the NMDR channels that would presumably release ket from channel block reduces the inhibition of the NMDAR eEPSCs JUST AS WOULD BE EXPECTED IF KET WERE A VERY LONG LASTING CHANNEL BLOCKER.

Fig. 5 shows that an experiment that presumably would untrap ket from NMDA receptors at one hr results in no NMDAR eEPSC reduction at 24 hrs (Fig. 5 d-h), whereas an experiment that would presumably trap additional ket at 10 min leads to reduction in NMDAR eEPSCs at 24 hrs (Fig. 5 i-L), as would be expected if ket were a very long lasting channel blocker.

The authors have presented data to support their hypothesis of long term trapping in real brain and brain slices. Are other explanations possible? Other explanations are always possible in such complex systems, but if one wants to study native receptors in real cells in real brain or brain slices, then the experiments are by necessity more complex in design and interpretation.

Yes, experiments will have to be done in model systems down the road as suggested by Reviewer #3, but these are different types of experiments for a different study.

In light of the concerns of Reviewer #3 I would suggest that the authors be more tempered in their conclusions (and in the title as well) indicating that their findings are consistent with (support) an hypothesis of long term trapping and do not directly establish long term trapping.

Again, we are immensely grateful to reviewer 2 for his/her continuous strong support and appreciation of our work!!!

We have now changed the title to “Sustained Antidepressant Effect of Ketamine Potentially via NMDAR Trapping in Lateral Habenula”, and changed “establish a causal role” to “support a hypothesis” in discussion (Page15, Line 332).

Also, in the discussion The authors might also consider the concept of whether delayed diffusion [B. Katz et al. J Physiol. 1973 The binding of acetylcholine to receptors and its removal from the synaptic cleft] might or might not play a role. Basically, as KET is released from NMDARs and diffuses out of the synaptic clefts it could likely rebind to unbound NMDARs receptors multiple times greatly extending its effective duration of action. Hence, a binding time of a few hours might be extended to days of action, and observed single channel binding times could be effectively extended many fold in the synaptic cleft through delayed diffusion when not observed in single channel

Thank you for bringing up this important point, and the insightful suggestion in summary of the Fig. 3 result above. Accordingly, we have now added the following paragraph into discussion:

“Another factor that could possibly contribute to ketamine’s extended action time *in vivo* is delayed diffusion¹². Unlike transmitters which have transporter-based clearance mechanism, ketamine could likely rebind to unbound NMDARs receptors multiple times as it is released and diffuses out of the synaptic clefts or extracellular spaces where astrocytic endfeet tightly wrap around neurons¹³. Given the difference in wash off experiments between memantine and ketamine (Fig. 3), such lateral diffusion seems unlikely to be the major explanation for the sustained effects, but may nevertheless contribute to the extended action time.”

Statistics: In re-reading the paper I found some potential problems with the statistics in a few parts of some figures which I list below that need to be checked by the authors. Of course these concerns could be errors in my measurements and calculations.

A rule of thumb for SEM error bars and unpaired t-tests is that if the SEM error bars overlap or are separated by short distances for the two data sets to be compared, then the differences in means are unlikely to be significant. This can be seen in Fig. 2

where there needs to be gaps between the error bars of the compared data sets to obtain significance, and large gaps to obtain higher significance.

In Fig. 1 K the right most panel for ket blue the undrawn downward error bar would overlap with the upward error bar. I calculate a $P \sim 0.25$ (not significant) for this while the authors calculate a P of: $*** < 0.001$, highly significant. (I used a two tailed t -test) This needs to be checked, and of course I may have mis-calculated.

In Fig. 1 K ket middle panel for comparison of baseline and 0-5 min I get a P of about 0.04, not < 0.0001 as the authors get. The other 0 to 1 hr significance levels are also suspect.

In Fig. 1 L middle and right there appear to be similar problems suggesting that the actual P values may be considerably greater than the indicated P values.

We appreciate the reviewer's careful reading. The data on bursting spike frequency and bursts per min in Fig. 1k,l are not normally distributed, as verified by D' Agostino-Pearson test, Shapiro-Wik test and Kolmogorov-Smirnov test. Therefore we used nonparametric test (Two way ANOVA Multiple comparisons or Mann-Whitney test), instead of t test here. Compared with the traditional t test method, this nonparametric test focuses more on the distribution characteristics of data (including median and quartiles). As shown in the violin plots below, there is a clear difference between the ketamine group and the saline group. In addition, for the 0-1 h data, we used paired statistical methods. From the scatter plot of Extended Data Fig. 3, it can be seen that ketamine significantly suppressed bursting activity within 1 hour. The P values calculated from these statistical methods are indeed low. Such detailed statistical methods for all the data analysis are now listed in the supplementary table1.

Figure R1. Single injection of ketamine causes prolonged suppression of LHb bursting activity *in vivo*.

a, b, Violin plots illustrating the bursting spike frequency (a) and bursts per minute (b) in CRS mice at different time points after saline or ketamine i.p. injection. $n = 46$ units in 4 mice in saline group and 51 units in 5 mice in ketamine group for 0-1 h data; $n = 72, 86$

units in 7 mice in saline group and 89, 92 units in 10 mice in ketamine group for 24 h data. *** $P < 0.001$; **** $P < 0.0001$; NS, not significant.

At the end of the Fig. 1 legend, the “n = 72, 86 units . . . and 89, 92 units” needs to be explained or corrected in terms of n and units. . Why are n and units values different?

Thank you for pointing out the need for clarification. As described in Method, “Spiking signals were continuously recorded for 1 h after ketamine treatment (10 mg kg⁻¹, i.p.) with headstage un-removed. For data collected at 24 h or 3 d after ketamine treatment, since animals need to return to homecage for rest, headstage was removed and remounted”. The units cannot be guaranteed to be identical before and after the remounting, and the number of units may also change. Therefore, we used the paired statistical method for the 0-1 h data, and unpaired statistical method for the 24 h and 3 d data. We have added this detailed explanation in Methods (Page39, Line 1111-1116).

Fig. 4b. For plot to the left of Fig. 4b, state directly if the error bars in this plot come from the 5 and 6 separate cells mentioned for n. State if there is only one 60 minute run for each cell. In Fig. 4b the observed plotted points do not appear to match the stated n's unless there is perfect overlap of some plotted points. If so move overlapping points laterally so they can be seen. Place the SEM bars in Fig. 4b as this is stated as an unpaired t-test. Check the P values in Fig. 4b.

Yes, the error bars in Fig. 4b come from the 5 and 6 separate cells mentioned for n and only one 60 minutes run for each cell. There are indeed coincidental overlaps of points and we have now moved the overlapping points laterally. We have also revised similar issues in Fig. 3b. SEM bars in Fig. 4b are now added.

ALL FIGURES NEED SEM BARS if t-tests are applied to the data and all P values need to be checked if the P values do not appear consistent with the SEM bars.

We have added SEM bars in Fig.2d,j,f,l,h,n, Fig.3b, Fig.4b, Fig.5g,k,Extended Data Fig.4d-f, Extended Date Fig.6c,e, Extended Data Fig.9e-g and checked all P values. Due to the limitation of legend length, we have now moved all statistical related n values, P values, and methods in legends into the supplementary Table 1.

Extended data Figure 3 d. The Fig. legend mentions SEM error bars but there are no error bars in this figure. State the test for part d.

We have now deleted the description of “SEM error bars” and added the test methods (Chi-Square test).

- 1 Lavé T, L. O., Poulin P, Parrott N. in *Applications of Pharmacokinetic Principles in Drug Development* (ed Krishna R) p. 133-175 (2004).
- 2 Nair, A. B. & Jacob, S. A simple practice guide for dose conversion between animals and human. *J Basic Clin Pharm* **7**, 27-31, doi:10.4103/0976-0105.177703 (2016).
- 3 J. IDVALL, I. A., K. F. ARONSEN AND P. STENBERG. KETAMINE INFUSIONS: PHARMACOKINETICS AND CLINICAL EFFECTS. *Br.J. Anaesth.* **51**, 1167 (1979).
- 4 Erhardt, W., Hebestedt, A., Aschenbrenner, G., Pichotka, B. & Blumel, G. A Comparative-Study with Various Anesthetics in Mice (Pentobarbital, Ketamine-Xylazine, Carfentanyl-Etomidate). *Res Exp Med* **184**, 159-169, doi:10.1007/Bf01852390 (1984).
- 5 Sato, Y. *et al.* Chronopharmacological studies of ketamine in normal and NMDA epsilon1 receptor knockout mice. *Br J Anaesth* **92**, 859-864, doi:10.1093/bja/ae144 (2004).
- 6 Zanos, P. *et al.* NMDAR inhibition-independent antidepressant actions of ketamine metabolites. *Nature* **533**, 481-486, doi:10.1038/nature17998 (2016).
- 7 Yang, Y. *et al.* Ketamine blocks bursting in the lateral habenula to rapidly relieve depression. *Nature* **554**, 317-322, doi:10.1038/nature25509 (2018).
- 8 Moda-Sava, R. N. *et al.* Sustained rescue of prefrontal circuit dysfunction by antidepressant-induced spine formation. *Science* **364**, doi:10.1126/science.aat8078 (2019).

- 9 Yang, C. *et al.* AMPA Receptor Activation-Independent Antidepressant Actions of Ketamine Metabolite (S)-Norketamine. *Biol Psychiatry* **84**, 591-600, doi:10.1016/j.biopsych.2018.05.007 (2018).
- 10 Beconi MG, H. D., Park L, Lyons K, Giuliano J, Dominguez C, Munoz-Sanjuan I, Pacifici R. Pharmacokinetics of memantine in rats and mice. *PLOS Currents Huntington Disease* (2012).
- 11 Mealing, G. A., Lanthorn, T. H., Murray, C. L., Small, D. L. & Morley, P. Differences in degree of trapping of low-affinity uncompetitive N-methyl-D-aspartic acid receptor antagonists with similar kinetics of block. *J Pharmacol Exp Ther* **288**, 204-210 (1999).
- 12 Katz, B. & Miledi, R. The binding of acetylcholine to receptors and its removal from the synaptic cleft. *J Physiol* **231**, 549-574, doi:10.1113/jphysiol.1973.sp010248 (1973).
- 13 Cui, Y. *et al.* Astroglial Kir4.1 in the lateral habenula drives neuronal bursts in depression. *Nature* **554**, 323-327, doi:10.1038/nature25752 (2018).

Reviewer Reports on the Second Revision:

Referees' comments:

Referee #2 (Remarks to the Author):

A-H have been addressed in previous reviews.

The authors have adequately addressed my additional review comments.